# Noisy metabolism can promote microbial cross-feeding

Jaime G Lopez[1], Ned S Wingreen[2]*

[1]Lewis-Sigler Institute for Integrative Genomics, Princeton University, Princeton, United States; [2]Department of Molecular Biology, Princeton University, Princeton, United States

**Abstract** Cross-feeding, the exchange of nutrients between organisms, is ubiquitous in microbial communities. Despite its importance in natural and engineered microbial systems, our understanding of how inter-species cross-feeding arises is incomplete, with existing theories limited to specific scenarios. Here, we introduce a novel theory for the emergence of such cross-feeding, which we term noise-averaging cooperation (NAC). NAC is based on the idea that, due to their small size, bacteria are prone to noisy regulation of metabolism which limits their growth rate. To compensate, related bacteria can share metabolites with each other to 'average out' noise and improve their collective growth. According to the Black Queen Hypothesis, this metabolite sharing among kin, a form of 'leakage', then allows for the evolution of metabolic interdependencies among species including de novo speciation via gene deletions. We first characterize NAC in a simple ecological model of cell metabolism, showing that metabolite leakage can in principle substantially increase growth rate in a community context. Next, we develop a generalized framework for estimating the potential benefits of NAC among real bacteria. Using single-cell protein abundance data, we predict that bacteria suffer from substantial noise-driven growth inefficiencies, and may therefore benefit from NAC. We then discuss potential evolutionary pathways for the emergence of NAC. Finally, we review existing evidence for NAC and outline potential experimental approaches to detect NAC in microbial communities.

*For correspondence:
wingreen@princeton.edu

**Competing interest:** The authors declare that no competing interests exist.

## Editor's evaluation

In this article, authors propose a novel hypothesis that can help explain why microbes release metabolites. In their NAC (noise-averaging cooperation) hypothesis, within-population cross-feeding can arise due to noisy metabolism in microbes. The authors predict substantial noise-driven growth inefficiencies from single-cell protein abundance data, review evidence for NAC, and propose how to detect NAC in microbial populations.

## Introduction

Microbial communities are found nearly everywhere in nature, inhabiting ecosystems ranging from hydrothermal vents (*Ding et al., 2017*) to mammalian guts (*Lloyd-Price et al., 2016*). One of the most striking properties of these communities is the ubiquity of cooperation: microbes frequently share resources, relying on other microbes for the acquisition of essential metabolites. This exchange of resources is broadly referred to as 'cross-feeding' and can involve a wide variety of compounds, ranging from organic acids to vitamins. Cross-feeding is widespread across both natural and engineered systems, with notable examples occurring in the human gut and in wastewater treatment systems (*Smith et al., 2019b*; *Nelson et al., 2011*). These metabolic interactions link organisms across

the entire tree of life, occurring both within and between kingdoms (*Sokolovskaya et al., 2020*) and even between specialized microbes of the same species (*Rosenthal et al., 2018*).

Cross-feeding plays a major role in the structure and function of microbial communities. In natural settings, cross-feeding is known to be a significant driver of microbial diversity, allowing many species to coexist on a small number of primary resources (*Goyal and Maslov, 2018*; *Goldford et al., 2018*). This microbial diversity has been linked to a wide variety of community properties (*Ptacnik et al., 2008*; *van Elsas et al., 2012*), including influence on host fitness (*Taur et al., 2014*). Cross-feeding can even play a role in public health: it has been shown that metabolite exchange can allow pathogens to compensate for fitness losses associated with antibiotic resistance (*Hammer et al., 2014*). In engineered systems, cross-feeding can be necessary for efficient operation. For example, wastewater treatment reactors rely on cross-feeding to prevent the buildup of inhibitory waste products (*Ciotola et al., 2014*). Thus, a thorough accounting of the factors promoting cross-feeding is an important part of understanding both natural and engineered microbial communities.

As a result of cross-feeding's key role in microbial communities, much work has been dedicated to unraveling its origin. The evolution of cross-feeding requires the emergence of two organisms: the organism secreting the metabolite and the organism consuming it. In some cases, the secreted metabolite could be a waste product that is costless or even beneficial to secrete (*Pfeiffer and Bonhoeffer, 2004*; *Pacheco et al., 2019*; *Kreft et al., 2020*). Thus, the secreting organism evolves as a consequence of metabolic optimization, while the resulting availability of the secreted metabolite provides an opportunity for the emergence of a consumer. However, many cases of cross-feeding involve the secretion of metabolites that are essential to both the secreting and consuming organism (*Sokolovskaya et al., 2020*). Why would an organism leak metabolites necessary for its own growth, and why would the consuming organism lose its capacity to produce an essential metabolite? The most popular theory for the evolution of this form of cross-feeding is the Black Queen Hypothesis (BQH), which focuses on the evolution of the consuming organism. The BQH posits that if there exists an organism that secretes a useful metabolite, there will be a selective advantage for some organisms to lose the ability to produce the metabolite and rely on leakage from the secreter (*Morris et al., 2012*; *Smith et al., 2019a*; *Morris et al., 2014*; *Sachs and Hollowell, 2012*). An alternative theory, the 'economies of scale' hypothesis, posits that cross-feeding is stabilized because organisms somehow gain an intrinsic fitness advantage from specializing their metabolite production and sharing with other such specialized organisms (*Pande et al., 2014*). Both of these hypotheses in principle provide theoretically sound mechanisms for consumer evolution and cross-feeding stabilization, but leave open the question of how leakage of essential metabolites first emerges. Thus, a complementary theory is needed that can explain the emergence of leakers.

It has been posited that some cross-fed metabolites are naturally leaky (*Morris et al., 2012*), but this assumption is not well-supported. For a limited set of functions, leakage is clearly unavoidable because key processes take place outside the cell, such as the hydrolysis of large polymers by extracellular enzymes (*Gore et al., 2009*). However, many cross-feeding relationships involve metabolites that are produced intracellularly, and it is generally not known how these metabolites exit the cell, much less that this leakage is inevitable. Polar or charged metabolites are known to have low membrane permeability, limiting the possibility of natural leakage through the cell membrane (*Chakrabarti and Deamer, 1992*). Indeed, even if a metabolite has a high membrane permeability, this does not necessarily indicate high absolute leakage rates. Cells could minimize leakage by maintaining only a small metabolite pool with rapid turn-over or by storing the metabolite in an altered, less leakage-prone form. Even if substantial quantities of a metabolite are observed to escape via a leaky membrane, it cannot be determined without further study whether this leakage is truly inevitable or, rather, advantageous in some way. If leakage isn't inevitable, how might it emerge?

Here, we explore a novel cooperative behavior that promotes the emergence of leakiness and thus cross-feeding in microbial communities. The basis of this mechanism is that metabolic enzyme regulation has been shown to be noisy, particularly so in bacteria due to their small size (*Taniguchi et al., 2010*). The resulting imbalances in enzyme levels can lead to growth inefficiencies due to the under or overproduction of necessary metabolites. Rather than attempting to downregulate the activity of the excess enzymes, cells can in principle improve their communal growth rate by exchanging metabolites among their kin and effectively 'averaging' out intracellular noise. We term this mechanism noise-averaging cooperation (NAC). The emergence of NAC among identical individuals creates the

conditions for obligate consumers to evolve via gene deletions, as proposed by the BQH. Thus, the proposed mechanism provides a complementary theory in which metabolic leakiness is not assumed a priori, but rather arises from selective pressures. We first characterize NAC in an ecological model of a small population of cells, demonstrating that metabolite leakage can increase collective fitness. We find that in extreme cases, NAC can even prevent the death of cells whose poor enzyme regulation would otherwise lead to irreversible growth arrest. We then develop a generalized, experimentally accessible framework for estimating how NAC is influenced by community size and the complexity of metabolic pathways. Using this framework and single-molecule data on *Escherichia coli* enzyme levels, we predict that typical bacteria suffer from significant growth inefficiencies due to imperfect regulation, and thus could benefit from metabolite exchange. We then show that the benefits of NAC can be privatized, a key requirement for the evolutionary accessibly of cooperative behaviors. Finally, we discuss potential experimental tests of the theory.

## Results

### Isolated cells

We begin by exploring the impact of enzyme noise on the growth of an isolated model cell. To focus on the role of enzyme level fluctuations, we consider a cell of fixed volume and track the numbers of internal enzyme and metabolite molecules. Cell growth rate is recorded, but does not explicitly lead to an increase in cell volume. Instead, to capture the effects of cell growth and the associated volume increase, the rates of enzyme production and enzyme loss by dilution are both taken to be proportional to the growth rate. Since metabolite production and consumption fluxes are generally large compared to the dilution of metabolite levels by growth, we neglect the small effect of dilution on the metabolite levels. In this simple model, we assume that cell growth requires two essential metabolites that have intracellular counts of $m_1^{\text{int}}$ and $m_2^{\text{int}}$. These metabolites are produced intracellularly by specialized enzymes with intracellular counts of $E_1$ and $E_2$. Both metabolites are produced from the same precursor, which is imported such that a constant number of precursor molecules is maintained within the cell. Each enzyme produces its metabolite at rate $\kappa E_i$, with $\kappa$ encompassing both the precursor concentration and the enzyme rate constant. For simplicity, we neglect the role of posttranslational modification in regulating enzyme activity (see Discussion for details on the potential impact of such additional regulation). In accordance with Liebig's law of the minimum, growth is proportional to the level of the least abundant of the two metabolites such that the growth rate is $g = g^* \text{Min}_i(m_i^{\text{int}})$, where $g^*$ is a constant relating the metabolite levels to the cell growth rate. Thus, the cell growth rate is maximized when the metabolites are produced and present in equal amounts. Metabolites within the cell are assumed to be consumed at a rate proportional to the growth rate. Metabolites are also exchanged with an extracellular space whose volume is $r_{\text{V}}$ times greater than a cell volume. This exchange occurs via membrane diffusion with permeability $P$ (we show in Appendix 1 that this form of exchange is mathematically equivalent to active transport in the linear regime). Metabolites both inside and outside the cell are passively degraded at a rate $\delta$. A schematic of this model is shown in *Figure 1A*. Formally, the intra- and extracellular counts of the metabolites evolve according to the following time-rescaled equations (see Appendix 1 for details):

$$\frac{dm_i^{\text{int}}}{dt} = \kappa E_i - \text{Min}_j(m_j^{\text{int}}) + P \cdot (m_i^{\text{ext}}/r_{\text{V}} - m_i^{\text{int}}) - \delta m_i^{\text{int}}, \tag{1}$$

$$\frac{dm_i^{\text{ext}}}{dt} = -P \cdot (m_i^{\text{ext}}/r_{\text{V}} - m_i^{\text{int}}) - \delta m_i^{\text{ext}}. \tag{2}$$

We model these metabolite dynamics as deterministic, as there is generally a large number of each essential metabolite within cells (*Bennett et al., 2009*).

Enzyme production within the cell is regulated based on internal metabolite levels, with the cell exclusively producing the enzyme corresponding to the currently least abundant metabolite. A flowchart of this regulation scheme is shown in *Figure 1B*. To reflect the bursty nature of gene expression (*Golding et al., 2005*), we assume that enzymes are produced in Poisson distributed bursts with average size $\beta$. Cells produce enzymes at a rate proportional to their growth rate such that the rate of enzyme bursts is $(\gamma/\beta)g^* \text{Min}_i(m_i^{\text{int}})$, where $\gamma$ is a constant controlling the steady-state abundance of enzymes (such that at steady state $\langle E_i \rangle = \gamma/2$). Enzymes are diluted by growth at a rate proportional

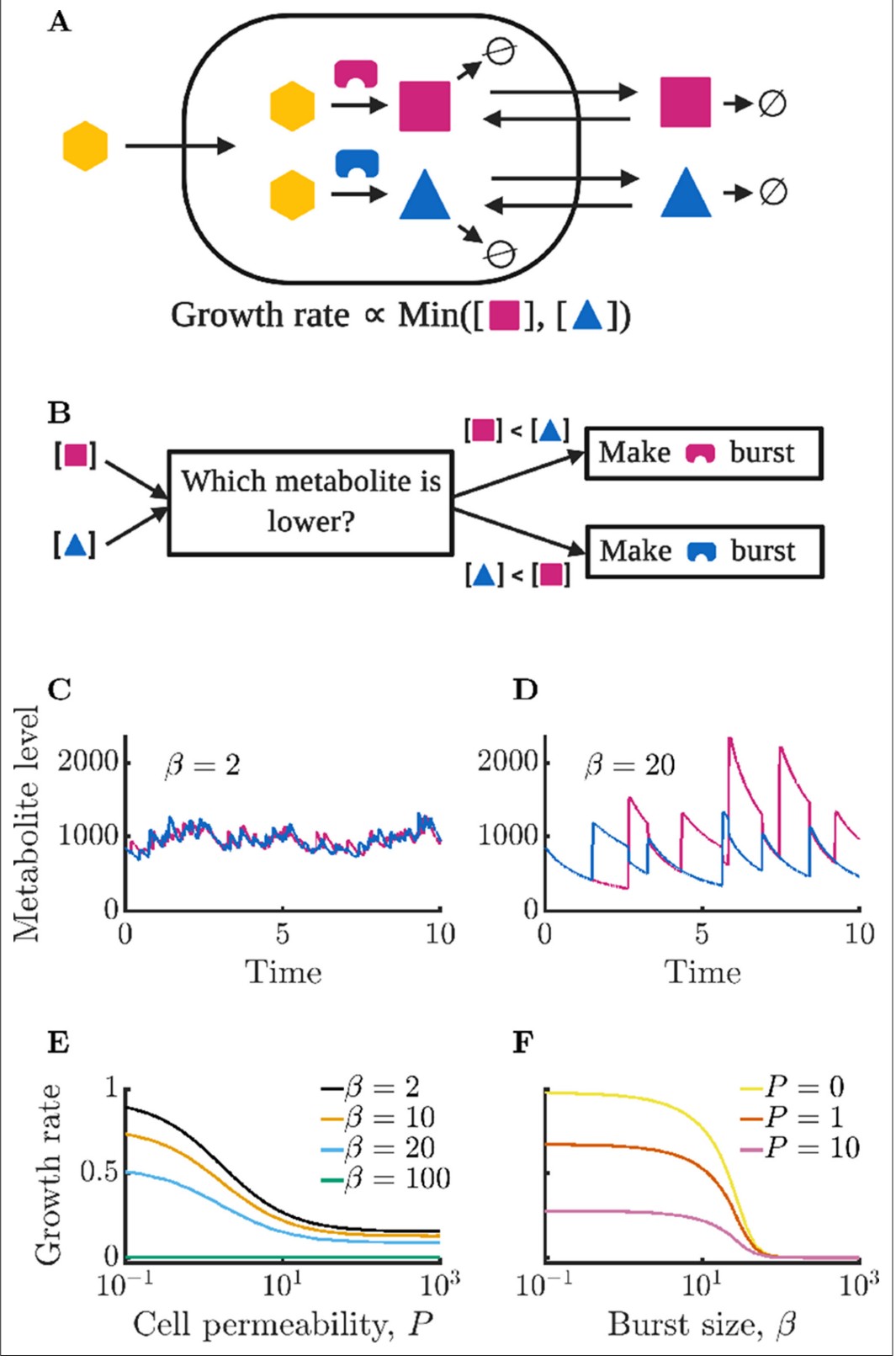

**Figure 1.** Isolated bacterial cells suffer negative growth effects from noisy enzyme regulation and metabolite leakage. (**A**) Schematic of modeled intracellular dynamics. Cells import an external nutrient (yellow hexagon) that can be converted by enzymes (magenta and blue) into two essential metabolites. Metabolites passively exchange with the extracellular medium ('leakage'), and degrade at a fixed rate. The two metabolites are used for growth

*Figure 1 continued on next page*

*Figure 1 continued*

in accord with Liebig's law of the minimum. (**B**) Schematic of dynamic enzyme regulation scheme: the type of enzyme produced is always the one associated with the lower metabolite pool. (**C**) Metabolite timecourse of a cell that produces enzymes in small bursts. See *Equations 1; 2* for details; parameters $\beta = 2$, $\gamma = 50$, $\kappa = 100$, $r_V = 10$, $P = 1$, $\delta = 1$, $g^* = 1 \times 10^{-5}$. Time is normalized by the inverse of the maximum steady-state growth rate ($t' = \frac{t\kappa\gamma g^*}{2(1+\delta)}$). (**D**) Metabolite timecourse of a cell that produces enzymes in large bursts. Same parameters as in *C* but with $\beta = 20$. (**E**) Average growth rate of an isolated cell for differing values of permeability; parameters as in *C* and *D* except as specified. Growth rate is normalized to the maximum possible growth rate, i.e. with perfect regulation and zero permeability. (**F**) Average growth rate of an isolated cell for differing values of burst size. Parameters as in *C* and *D* except as specified.

to their abundance $E_i g^* \mathrm{Min}_i(m_i^{\mathrm{int}})$. The coupling of enzyme production and dilution to growth rate reflects the requirement that all cellular components must increase on average at the growth rate to prevent component imbalances. We model enzyme production as a stochastic process due to its intrinsically noisy nature, and model enzyme dilution as a deterministic process. The metabolite and enzyme equations are simulated numerically using a hybrid deterministic-stochastic method (see Appendix 2 for details).

In *Figure 1C and D*, we show example timecourses of metabolite dynamics for different burst sizes in cells with an average level of each enzyme of $\langle E_i \rangle = \gamma/2 = 25$. In *Figure 1C*, we show the metabolite levels of a cell with a small burst size $\beta = 2$. The small bursts allow for precise regulation of metabolite production, with the cell maintaining nearly equal levels of the two metabolites. In contrast, *Figure 1D* shows a cell with a burst size of $\beta = 20$. This cell's poor enzyme regulation leads to imbalances in enzyme levels which in turn manifest as metabolite imbalances (see *Appendix 9—figure 1* for the corresponding enzyme timecourses).

How does the growth of an isolated cell depend on membrane permeability? In *Figure 1E*, we show the mean growth rate of isolated cells for varying permeability $P$. As can be seen, growth rate decreases monotonically with permeability. This follows because permeability leads to a loss of metabolites to the extracellular space where they cannot be utilized by the cell, but can be degraded. The coefficient of variation (CV) of the intracellular metabolite levels does not substantially change with increasing permeability, decreasing only very slightly due to stored metabolites in the extracellular space buffering fluctuations within the cell (see *Appendix 9—figure 2*). Growth rate losses increase with growing extracellular volume, approaching the limit in which metabolites are permanently lost upon leakage from the cell (see *Appendix 9—figure 3*).

The growth of isolated cells is also strongly influenced by the enzyme burst size, with small burst sizes permitting faster growth. This is seen in *Figure 1F*, which shows that growth rate decreases monotonically with average burst size $\beta$. The decreasing trend reflects a type of 'use it or lose it' phenomenon in which cells grow poorly when they have large metabolite imbalances, as these result in metabolites being degraded instead of consumed for growth. The smaller the burst size, the lower the variance of the enzyme levels and the closer to equality metabolite production and levels can be maintained. Note that with sufficiently large burst sizes, the cell can experience irreversible metabolic arrest. This occurs because the cell must grow to produce additional enzymes, and if the cell experiences a sufficiently large metabolite imbalance, it may be unable to make another burst of enzyme before its existing metabolites are exhausted. In *Figure 1E*, this growth arrest is reflected in the $\beta = 100$ curve that is near zero for all values of permeability. Similarly, in *Figure 1F*, there is a value of $\beta$ beyond which cells do not grow.

## Multi-cell communities

We have characterized the behavior of an isolated model cell, but how do enzyme noise and metabolite leakage affect growth rates in a community? We now expand our model to a population of cells that share a common extracellular space with which they exchange metabolites, as depicted in *Figure 2A*. When cells leak, there is now a possibility that these leaked metabolites will be taken up by other cells. To explore how leakage influences the collective metabolism of a multi-cell community, we simulate a community of 10 cells growing under the same conditions as in *Figure 1E*. In *Figure 2B*, we show the intracellular metabolite CV of these cells as a function of permeability for a range of enzyme burst sizes. As in the single-cell case, metabolite CV generally increases with increasing burst size.

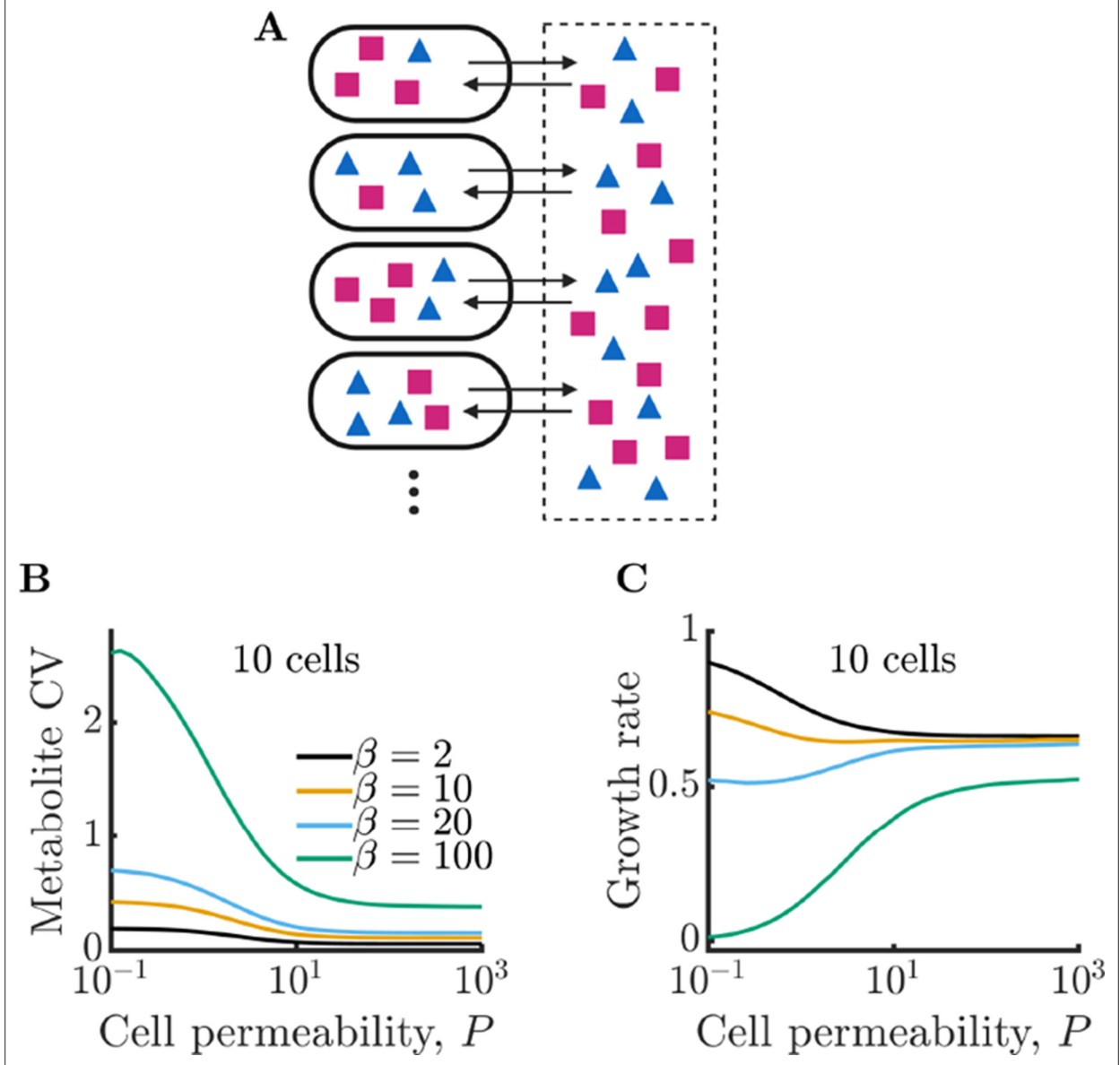

**Figure 2.** Bacterial cells can compensate for noisy enzyme regulation and increase growth rate by exchanging metabolites within a clonal community.
(**A**) Schematic of multi-cell metabolism model. Individual cells regulate their own enzyme levels, but metabolites leak into the local medium and can be used by other cells in the community. (**B**) Intracellular metabolite coefficient of variation (CV) for a community of 10 cells as a function of cell permeability. (**C**) Average growth rate for community of 10 cells as a function of cell permeability. Parameters in *B* and *C* same as in *Figure 1E*.

Interestingly, however, the metabolite CV now decreases substantially with increasing permeability. This occurs because metabolite exchange allows the cells to 'average out' the noise arising from their individually poor enzyme regulation, a phenomenon we term noise-averaging cooperation (NAC). With large burst sizes, cells are prone to overproducing one type of enzyme, and thus overproducing a single type of metabolite. An isolated cell has no avenue to remedy this imbalance, leading to degradation of the overabundant metabolite. In a sufficiently large community this changes: within the population of cells, it is likely that there exist cells with opposite imbalances, and by exchanging metabolites these cells can collectively balance their metabolism.

How does this decrease in metabolite noise impact the average growth rate of cells within the community? In *Figure 2C*, we plot the growth rates of the communities from *Figure 2B*. In the case of a large burst size ($\beta = 100$), the improvement is extreme. With sufficiently high permeability, cells that were previously unable to grow at all due to their poor regulation can now grow at a substantial

fraction of the optimal growth rate. We note that while metabolite CV decreases for all values of burst size, this does not always translate into a growth improvement. With very small burst sizes ($\beta = 2$), the increased permeability has the opposite effect and slightly decreases the growth rate. Since these cells already have efficient enzyme regulation, the moderate decrease in metabolite CV is outweighed by the increased degradation of metabolites within the extracellular space. Thus, the benefits of NAC are greatest under two conditions: (1) when individual cells have poor enzyme regulation and (2) when cells exist in a crowded space with minimal free volume between cells (such as in a biofilm).

How does community size influence the noise-reducing effect of NAC? To answer this question, we consider a simplified, linear version of our model in which a population of $n$ fully permeable cells are directly connected to each other. We track only a single type of metabolite and enzyme. We also remove the nonlinear feedback between growth and enzyme dynamics, assuming a constant growth rate such that the per-cell average rate of enzyme bursts is $\Gamma/\beta$ and the rate of enzyme dilution is $\mu_E$. Metabolite consumption and degradation are aggregated into a single rate parameter $\mu_m$. For simplicity, we assume that all enzyme bursts are of size $\beta$, rather than being Poisson distributed. The Langevin equations for the total number of enzymes and total number of metabolites are therefore:

$$\frac{dE}{dt} = n\Gamma - \mu_E E + \xi_E(t), \tag{3}$$

$$\frac{dm}{dt} = \kappa E - \mu_m m + \xi_m(t), \tag{4}$$

where the $\xi(t)$ are noise terms with $\langle \xi(t) \rangle = 0$ and noise strengths $\langle \xi_E^2(t)\delta t \rangle = n\Gamma(\beta + 1)$ and $\langle \xi_m^2(t)\delta t \rangle = 2\kappa\Gamma n/\mu_E$ (see Appendix 3 for further details). Since these equations are linear, we can exactly compute the expression for the CV of the total enzyme level (see Appendix 3):

$$\mathrm{CV}_E = \sqrt{(\beta + 1)\left(\frac{\mu_E}{2\Gamma n}\right)}. \tag{5}$$

Consistent with our simulations, the CV increases with burst size $\beta$. The dependence on population size can also be immediately seen from this expression, with the CV being proportional to $1/\sqrt{n}$. Thus, larger communities are expected to magnify the positive impact of metabolite exchange. We can also directly compute the metabolite CV:

$$\mathrm{CV}_m = \sqrt{\frac{\mu_E \mu_m(\kappa\beta + 2\mu_E + 2\mu_m + \kappa)}{2\kappa\Gamma n(\mu_E + \mu_m)}}. \tag{6}$$

As expected, the metabolite CV has a similar scaling with respect to $n$ and $\beta$ as the enzyme CV. In (*Appendix 9—figure 4*), we compare these predicted relationships to the full model and find good agreement.

These calculations, along with our simulations, characterize the potential benefits of NAC. The small size of bacteria make them inevitably noisy, possibly leading to growth losses. Metabolite leakage can act as a form of bacterial mutual aid, benefiting cells by allowing resource pooling.

## Generalized NAC framework

We have demonstrated NAC in a simple model with two metabolites, but how would the benefits apply to more realistic metabolic networks? We now develop a general framework to determine the impact of enzyme fluctuations and metabolite sharing on the growth of bacteria with an arbitrary number of non-substitutable metabolites. We begin with an arbitrary probability distribution function (PDF) of the intracellular levels of an individual metabolite $f(m_i)$. For simplicity, we first consider the case of independent but otherwise identical metabolites. For a given number of non-substitutable metabolites, we can then apply Liebig's law of the minimum to compute the distribution of growth rates $q_n(g)$ by determining the distribution of the lowest metabolite level among a set of $n$ metabolites. Note that while this minimum is technically only proportional to growth rate, for brevity we assume $g^* = 1$ such that $g = \mathrm{Min}_i(m_i)$. To calculate $q_n(g)$, we use the cumulative distribution functions (CDFs) $F(m_i)$ and $Q_n(g)$:

$$q_n(g) = \frac{dQ_n(g)}{dg} = \frac{d}{dg}\left\{1 - [1 - F(g)]^n\right\}. \tag{7}$$

Intuitively, the mean of the distribution of growth rates with $n > 1$ non-substitutable metabolites will always be lower than the mean of $m_i$, as shown schematically in *Figure 3A*. Thus, as in our simple

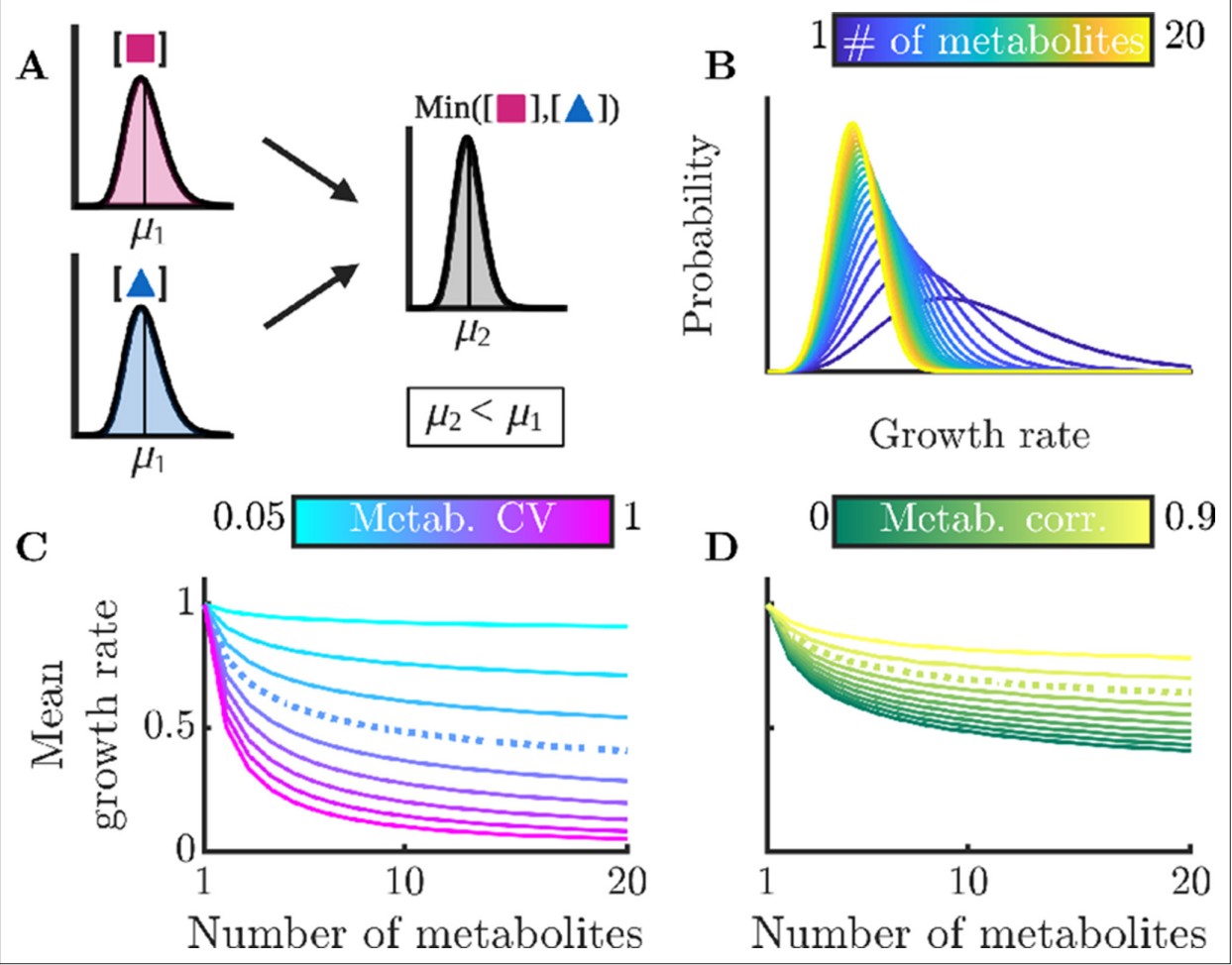

**Figure 3.** Sharing multiple metabolites can generically reduce noise and improve overall colony growth rate. (**A**) Implication of Liebig's law of the minimum for fluctuating metabolites: the growth rate at any time is set by the lowest metabolite level, hence the average growth rate is lower than at the average metabolite level. The magnitude of this decrease grows with increasing metabolite variance. (**B**) Distributions of growth rates, set by minimum metabolite level $\mathbf{Min}(m_i)$, for varying numbers of essential metabolites. (**C**) Mean growth rate as a function of number of metabolites and of metabolite CV. The dashed curve corresponds to CV =0.4, as measured for essential proteins in *E. coli* (***Taniguchi et al., 2010***). (**D**) Mean growth rate as a function of number of metabolites and of the correlation coefficient between metabolites, for CV=0.4. The dashed curve indicates a metabolite correlation of 0.7, approximately that observed for proteins in *E. coli* (***Taniguchi et al., 2010***).

two-metabolite model, the mean growth rate will depend not just on the means of the individual metabolite counts, but also their variability.

To examine this general model in more detail, we now use realistic metabolite distributions to quantitatively analyze the effect of various factors on the growth-rate distribution. While the distribution of metabolite levels in single cells has not been directly measured, there have been extensive measurements of single-cell protein distributions showing that these levels are typically gamma distributed. As we expect that enzyme levels are the dominant source of metabolite noise, we therefore approximate the metabolite distributions as gamma distributions, that is we assume that the metabolite distributions inherit the shape and thus the CV of the underlying enzyme distribution. In particular, we use the median gamma-distribution parameters measured for essential proteins in *E. coli* (***Taniguchi et al., 2010***) as a base. (Note that we consider the measured variability to be due to stochastic noise, but in principle some of this variability could emerge from deterministic dynamics, e.g. persistent oscillations. This distinction is not important for NAC, as fluctuations away from an optimal average arising from deterministic dynamics will have the same effect as stochastic noise.) For simplicity, these analyses assume that all metabolites have the same average concentration and are thus co-rate limiting. We show the corresponding growth-rate distribution as a function of the

number of non-substitutable metabolites in *Figure 3B*. One sees that as the number of metabolites increases, the mean of the growth-rate distribution decreases. This occurs because the more metabolites the cell must manage, the more likely it is that at least one will be poorly regulated at a given moment and constrain growth. Thus, NAC is most beneficial to cells that require large numbers of non-substitutable metabolites. In *Figure 3C*, we explore how this decrease in growth rate depends on the CV of the metabolite distribution, with the CV of the *E. coli* essential protein distribution shown as the dashed curve. If the metabolite levels are poorly controlled resulting in a large CV, the addition of more metabolites drastically reduces growth. Conversely, if the cell has tight control of its metabolites, it can manage significant numbers of non-substitutable metabolites without too large a growth loss. Interestingly, the curve corresponding to the CV of essential *E. coli* proteins shows a significant loss in growth rate, suggesting that real-world bacteria may suffer substantially from poor enzyme regulation. It should be noted, however, that enzyme count noise may overestimate noise in the resulting metabolites, as the enzymes can be regulated post-translationally, and metabolite fluxes may be buffered against enzyme fluctuations by network feedback effects (*Hackett et al., 2016*).

Thus far, we have assumed that the metabolites' distributions are independent, but this likely does not hold in nature. Experimental studies have found that levels of different enzymes within the cell are correlated (*Taniguchi et al., 2010*), suggesting that metabolite levels are likely also correlated. Indeed, this phenomenon occurs even in the simple models we analyzed above (see *Appendix 9—figure 1*). To account for these correlations, we computed the mean growth rate for varying degrees of correlation between metabolites. As a baseline, we again use the median distribution of essential proteins in *E. coli*. The results can be seen in *Figure 3D*, and show that correlation between metabolites reduces the adverse effects of metabolite noise. This occurs because if the metabolite levels are correlated, it is less likely that there will be a single outlying low metabolite level constraining growth. Note that the growth benefit of correlated metabolite fluctuations is largest when metabolites are on average co-rate limiting: if there is one metabolite that is strongly limiting, it makes little difference whether the other metabolites are correlated with it or not. The correlation between certain proteins in *E. coli* has been measured (*Taniguchi et al., 2010*), and we show this value as a dashed curve. While this level of metabolite correlation does improve growth, the growth loss associated with realistic enzyme noise is still substantial.

What if the growth rate is not determined by Liebig's law of the minimum? Real growth functions are unlikely to be quite so simple, and given the variation that exists in microbial metabolism, it is unlikely that there is a single universally applicable growth function. Despite this uncertainty, we can determine what classes of growth function will lead to noise-driven growth defects. Consider an arbitrary growth function $g(\mathbf{X})$ and vector of randomly varying metabolites $\mathbf{X}$. We can express the decrease in growth due to noise as $\mathbb{E}[g(\mathbf{X})] \leq g(\mathbb{E}[\mathbf{X}])$. This statement is equivalent to the multivariate Jensen's Inequality for concave functions, meaning that if $g(\mathbf{X})$ is concave, the introduction of metabolite noise will decrease the mean growth rate. Note that there are also functions which are not globally concave that also satisfy this inequality, see Appendix 5 for examples. Growth functions in which the benefit of increasing individual non-substitutable metabolite levels is saturating will generally be concave. Thus, most reasonable growth functions will lead to decreased mean growth in the presence of metabolite noise. To demonstrate this, in (*Appendix 9—figure 6*) we show a version of *Figure 3C* with an alternative growth function based on the rate of protein synthesis. Interestingly, the dependence of the noise-drive growth loss on the concavity of $g(\mathbf{X})$ implies that the magnitude of the loss may depend on the mean metabolite levels. If the metabolite levels are well above the saturation point of the growth process, such that the local growth function has low concavity, there will be minimal growth losses due to metabolic noise. A similar reduction in growth losses may occur if the metabolites are far below saturation.

With the above generalized framework, we were able to incorporate experimental measurements into our theory. Our preliminary analyses based on enzyme level measurements suggest that real bacteria may indeed suffer from substantial noise-driven growth defects. Combined with our analyses of simple metabolic models, this raises the possibility that bacteria can engage in NAC to improve their collective growth rate, particularly in tightly packed environments like biofilms.

## Discussion

In this work, we develop a theory of noise-averaging cooperation (NAC), a novel mechanism potentially underlying the emergence of both intraspecies cooperation and interspecies cross-feeding. NAC allows microbes with individually poor regulation to average out their metabolic noise and raise their collective growth rate by sharing metabolites. Since NAC is strongest in crowded environments, it suggests an additional benefit of the biofilm mode of growth. With respect to cross-feeding evolution, our mechanism can be viewed as complementary to the Black Queen Hypothesis: it provides a mechanistic explanation for the emergence of metabolite leakage, setting the stage for the evolution of metabolic interdependencies via gene deletions (*McNally and Borenstein, 2018*). However, while we have shown that NAC can be beneficial under plausible assumptions about metabolite noise and is consistent with some existing data, further study is needed to determine whether NAC is evolutionarily accessible and to what extent it may be a driver of cross-feeding in nature.

An important aspect of NAC to consider is its evolutionary stability. We have shown in this manuscript that NAC is a potential social optimum for densely packed bacteria. However, a state being socially optimal does not guarantee that the state is evolutionarily accessible or stable. In order for NAC to support the evolution of cross-feeding, cells engaging in NAC must be able to invade populations of cells not engaging in NAC and vice versa. The conditions for the maintenance of cooperative behaviors within a population have been studied extensively, and one of the key conditions is based on preferential interaction between 'cooperators'. If cooperators primarily interact with other cooperators, growth losses due to 'cheaters' are minimized (*Nowak, 2006*; *Nowak and May, 1992*). We conjecture that within a biofilm NAC cells will segregate with other NAC cells simply due to proximity within a lineage, providing a form of preferential interaction and stabilizing NAC within the population. Consider a growing biofilm composed of cells not engaging in NAC. If a cell engaging in NAC invades, its growth will initially be lower than its neighbors, due to the fact that its neighbors are only uptaking metabolites. However, after a few cell divisions, the single NAC performing cell will grow into a small sector of cells performing NAC and will thus largely share metabolites with its kin and potentially gain a growth advantage over non-NAC cells. To estimate this sharing distance, we developed a simplified model of NAC in a biofilm and parameterized it with measurements from a *Vibrio cholerae* biofilm (see Appendix 7). We found that leaked metabolites travel only a few cell layers before being largely consumed, suggesting that NAC cells require relatively few divisions to establish a successful NAC sector. Now consider invasion of a non-NAC cell into a NAC biofilm: a cell not engaging in NAC can invade because it is only taking up metabolites and not leaking them. However, its growth will eventually lead to a sector of cells not engaging in NAC, thus losing the benefit that allowed the initial invasion. Studying the evolutionary stability of NAC is a promising direction of future study, requiring the development of a spatial model incorporating cell growth along with metabolic noise.

If NAC does occur in nature, how can we identify it? Our theory predicts, counterintuitively, that it may benefit organisms to secrete essential metabolites into the environment. Thus, deliberate leakage or export of essential metabolites, such as amino acids or vitamins, is a potential signature of NAC. The clearest signature of deliberate export would be the existence of dedicated transporters for essential metabolites. Deliberate export could also occur via membrane leakage, but this case is more ambiguous as it is difficult to determine whether such leakage is 'deliberate', that is allowed by the cells, or is an unavoidable consequence of membrane permeability. As a test case, we examined the export of amino acids, a key class of non-substitutable metabolites, in *E. coli*. How much of *E. coli*'s amino-acid production is lost to leakage? Using prior experimental measurements of membrane permeability, intracellular concentrations, and amino-acid production rates, we estimate that *E. coli* loses only a small fraction (<1%) of its amino-acid production to membrane leakage (see *Appendix 9—figure 5*). This suggests that membrane leakage is not a substantial avenue of amino acid export in *E. coli*. On the other hand, there exist multiple amino-acid exporters in *E. coli* (*Pathania and Sardesai, 2015*; *Hori et al., 2011*; *Nandineni and Gowrishankar, 2004*; *Kutukova et al., 2005*; *Zakataeva et al., 1999*), implying that *E. coli* does indeed engage in deliberate export of essential metabolites, consistent with NAC. Moreover, consistent with the idea that these exporters enable metabolic exchange, it has been shown that artificial auxotrophic *E. coli* strains can indeed cross-feed each other amino acids (*Mee et al., 2014*). Note, however, that there may be reasons other than NAC for deliberate export of amino acids, such as overflow metabolism or the use of amino acids as signaling molecules (*Burkovski and Krämer, 2002*; *Park et al., 2003*; *Pathania and Sardesai, 2015*). Indeed, the 'economies of scale'

scenario, in which organisms are hypothesized directly benefit from specializing in production of fewer metabolites (*Pande et al., 2014*), might lead to similar amino acid exporting behavior.

Another way to probe the relevance of NAC to real bacteria would be to obtain more accurate estimates of intracellular metabolite distributions. Due to a lack of direct measurements of metabolite concentrations within single bacterial cells, we approximated the metabolite distribution using data from single-cell proteomics measurements. This approximation allowed us to estimate growth loss due to noise, but our conclusions depend on the assumption that the magnitude of metabolite noise roughly follows that of enzyme noise. In our simple metabolic model, this assumption is borne out, as the dominant source of metabolite noise is the burstiness of the enzyme dynamics. However, in real cells, there are likely additional regulatory feedbacks that suppress metabolite noise. Thus, our analysis may overestimate metabolite noise and thus the benefits of NAC. Future theoretical studies could yield a more accurate estimate of metabolite noise using realistic models of intracellular metabolite dynamics that incorporate complete pathways and phenomena such as post-translational regulation. There is also the possibility that direct measurements of intracellular metabolite distributions will become available, as the technology for single-cell metabolite measurement is rapidly advancing (*Evers et al., 2019*; *Yoshida et al., 2019*).

In addition to more precisely quantifying intracellular metabolite noise, better predicting noise-driven growth losses will require understanding the relationship between individual metabolite levels and growth rate in real cells. We showed that in order for metabolic noise to decrease mean growth rate, the only requirement is that the growth function be concave. We expect most growth functions to meet this condition, as the benefit of increasing metabolite levels generally saturates. However, it is possible that cells mitigate the impact of noise by maintaining their metabolite levels in a region of the growth function with low concavity, for example in a linear regime or near saturation. Intriguingly, one experimental study examining the relationship between fluctuations in metabolic enzymes and growth rates found that enzyme fluctuations did translate into growth fluctuations (*Kiviet et al., 2014*). These findings suggest that metabolite noise driven by enzyme fluctuations can indeed influence growth, as is required for NAC to be beneficial. However, more definitive conclusions can only be drawn from direct correlation of metabolite levels and growth rates, an experiment that is difficult with current technology. Such experiments may become possible in the future, as the technology for real-time single-cell metabolite measurement is rapidly developing. For example, a fluorescent reporter for branched-chain amino acids has recently been demonstrated in eukaryotes (*Yoshida et al., 2019*).

Even if the growth loss due to metabolic noise is large, for NAC to be beneficial the reduction in noise must outweigh the cost of metabolite loss in the extracellular space. In the context of a biofilm, some of this loss will likely arise from diffusion away from the biofilm, and thus the loss rate can potentially be calculated from the geometry of the biofilm and the diffusion constant of the metabolite within the biofilm matrix. Estimating the impact of other, spontaneous or reaction-based, forms of metabolite loss will likely require experimental measurements. We explore the impact of low degradation rates on NAC in Appendix 8.

Analysis of intracellular metabolite dynamics and realistic growth functions may provide some support for NAC, but definitive testing of the mechanism will likely require dedicated experiments. *E. coli* would be a suitable organism for such experiments, as it is known to encode transporters for at least some essential metabolites, and has already been shown to engage in intercellular exchange of amino acids (*Mee et al., 2014*). To directly test the theory, experiments will require at least two conditions: one in which cells are isolated and another in which they exist at a relatively high density. One possibility is to compare planktonic and biofilm cells, while another would be to assemble varying densities of planktonic cells. With isolated and crowded conditions defined, there are two major predictions that could be tested: growth rate should increase when cells are in crowded environments, and metabolic noise should be reduced when cells are in crowded environments.

Testing of the growth-rate prediction could be performed using population-level measurements of well-mixed cultures. The simplest way to test this would be measure the exponential-phase growth rates of bacterial cultures of different densities. An exponential-phase culture of *E. coli* could be resuspended at different densities in minimal media with saturating concentrations of nutrients, and growth rates measured (e.g. via OD). If NAC is occurring, the growth rate should be positively correlated with the culture density. Another possible experimental system is an *E. coli* chemostat fed with minimal media. The measured output in this system will be the steady-state biomass. NAC predicts that,

compared to the case where growth rate is independent of density, the cell density will be higher than expected at low dilution rates (see Appendix 6 for details). For both of the above experiments, it will be important to determine whether the observed growth differences are due to metabolite exchange. To test this, one could employ mutants with different essential metabolite exporter genes deleted and measure whether the difference in growth rate still exists between isolated and dense conditions. Data from these mutants will need to be interpreted carefully, for example due to redundant/undiscovered transporters or unintended effects of the deletions. In these experiments, NAC should be distinguishable from the 'economies of scale' mechanism of cross-feeding promotion mentioned earlier. In theory, the 'economies of scale' benefit should only occur when some organisms are auxotrophs for the shared metabolites, while NAC is beneficial even if all organisms are prototrophs.

Testing whether metabolic noise decreases with cell density will require techniques with single-cell resolution. The difficulty in testing this prediction will stem from finding a method to measure single-cell metabolite concentrations. One possible method to estimate the metabolite distribution is to measure timeseries of metabolite levels using the fluorescent reporter approach discussed above. One could image these fluorescent reporters in two-dimensional colonies and attempt to correlate the observed metabolic noise to local cell density. Similar to the earlier proposed experiments, one could employ export mutants to determine whether observed decreases in noise are due to metabolite exchange.

If NAC does exist in nature, how is it implemented and regulated by cells? It is unlikely that cells would exchange their entire metabolome with the external environment, but how would cells select which metabolites to exchange? If the metabolites within cells have different levels of noise, it might be optimal for cells to engage in NAC with the metabolites with the highest noise. This would imply that metabolites produced in small quantities are good candidates for NAC. The choice of which metabolites to exchange is also influenced by the architecture of the cell's metabolic network. If there is a bottleneck in the network, for example if catabolism generates a small set of metabolites that are then used in a myriad of anabolic synthesis processes, it may be beneficial to exchange these bottleneck metabolites. In addition to selecting which metabolites to exchange, there is also the issue of deciding when to engage in NAC. It would be harmful for cells to engage in NAC at low cell density, as most of the secreted metabolites would be lost. Thus, regulation of NAC would likely be linked to quorum sensing. Note, however, that sensing of kin cell densities would not be sufficient for regulation of NAC. If the environment also contains a high density of non-kin, there will likely be a high effective loss rate of metabolites to these non-kin, making NAC disadvantageous. Thus, regulation of NAC should be dependent on multiple quorum-sensing circuits, using both kin and non-kin autoinducers (*Miller and Bassler, 2001*).

Thus far, we have assumed that the level of noise within an isolated cell is immutable, but in nature metabolic noise is an evolvable trait. Thus, rather than engage in NAC, cells may have the option to increase growth by reducing intracellular noise. Substantial noise reduction is possible, but it is often costly with steeply diminishing returns. For example, one method of reducing noise originating from Poisson-type processes is to increase the number of the molecules of interest, which will decrease CV by the square root of the the number of molecules. This is essentially what occurs in NAC, but with production split across multiple cells. Another option is for the cells to introduce feedback-based regulation. This approach is even more costly, with information theoretic work showing that the CV of a molecule whose abundance is controlled by feedback only decreases with the fourth root of the number of signaling events (*Lestas et al., 2010*). These superlinear costs make it impossible for bacterial cells to be entirely noise-free given their size, but it may still be possible for a cell to eliminate most of the detrimental effects of noise. As a result, NAC must be considered as only one of multiple possible strategies for noise mitigation.

We have focused on bacteria in this manuscript, but it is also possible that NAC may be relevant in other domains of life. For example, NAC highlights a potential advantage of multicellularity: a multicellular tissue separated from its external environment is the optimal environment for NAC. NAC may also apply to macroecological systems if the outcome of foraging for resources is highly variable. Under such conditions, it may be beneficial for animals to engage in resource sharing, potentially supporting the development of social groups. Indeed, NAC is analogous to revenue-sharing clubs that can be employed to damp income volatility (*Tilman et al., 2018*).

While much research is needed to determine the relevance of NAC to real bacteria, the theory highlights an interesting aspect of ecology: noise at even the smallest scales can have a dramatic impact on the entire ecosystem. In this manuscript, we have focused on the single-species case, but there is potential for more novel behaviors in the many-species context, as has been observed in other resource-competition models (*Tikhonov and Monasson, 2017*). Overall, we hope this work can serve as a foundation for further theoretical and experimental work on how noise in resource acquisition impacts ecology and evolution.

## Methods

All code and data used in this manuscript can be found at https://github.com/jaimegelopez/NAC (*Lopez, 2021*, copy archived at swh:1:rev:d39206b0890340db8b4faad87ed544f345b09057). Details on individual analyses and derivations can be found in the relevant appendices.

## Acknowledgements

We thank Matt Black for useful discussions on possible experimental tests of the theory. JGL was supported by an NSF GRFP (DGE-1656466). This work was supported by the National Institutes of Health (R01 GM082938). This work was supported in part by the National Science Foundation, through the Center for the Physics of Biological Function (PHY-1734030).

---

## Additional information

### Funding

| Funder | Grant reference number | Author |
|---|---|---|
| National Science Foundation | DGE-1656466 | Jaime G Lopez |
| National Institutes of Health | R01 GM082938 | Ned S Wingreen |

The funders had no role in study design, data collection and interpretation, or the decision to submit the work for publication.

### Author contributions

Jaime G Lopez, Conceptualization, Data curation, Formal analysis, Investigation, Methodology, Validation, Visualization, Writing – original draft, Writing – review and editing; Ned S Wingreen, Conceptualization, Formal analysis, Funding acquisition, Investigation, Methodology, Supervision, Validation, Visualization, Writing – original draft, Writing – review and editing

### Author ORCIDs

Jaime G Lopez http://orcid.org/0000-0003-1647-5898
Ned S Wingreen http://orcid.org/0000-0001-7384-2821

### Decision letter and Author response

Decision letter https://doi.org/10.7554/eLife.70694.sa1
Author response https://doi.org/10.7554/eLife.70694.sa2

---

## Additional files

### Supplementary files

• Transparent reporting form

## Data availability

The current manuscript is a modeling study, and thus no data was generated for this manuscript. All modeling code is available at https://github.com/jaimegelopez/NAC, (copy archived at swh:1:rev:d39206b0890340db8b4faad87ed544f345b09057).

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

## Appendix 1

### Time-rescaling of the two-metabolite model

To show how we obtained the time-rescaled *Equations 1; 2*, we begin with a generalized set of equations in which a parameter $\alpha$ relates the minimum of the two metabolite levels to the consumption rate of the metabolites:

$$\frac{dm_i^{\mathrm{int}}}{dt'} = \kappa' E_i - \alpha \mathrm{Min}_j(m_j^{\mathrm{int}}) + P' \cdot (m_i^{\mathrm{ext}}/r_V - m_i^{\mathrm{int}}) - \delta' m_i^{\mathrm{int}}, \tag{8}$$

$$\frac{dm_i^{\mathrm{ext}}}{dt'} = -P' \cdot (m_i^{\mathrm{ext}}/r_V - m_i^{\mathrm{int}}) - \delta' m_i^{\mathrm{ext}}. \tag{9}$$

Here, $m_i^{\mathrm{int}}$ is the intracellular count of metabolite , $t'$ is time, $\kappa'$ relates enzyme count to metabolite production, $E_i$ is the count of enzyme , $\alpha$ relates the minimum of metabolite levels to metabolite consumption, $P'$ is permeability, $m_i^{\mathrm{ext}}$ is the extracellular count of metabolite , $r_V$ is the ratio of extracellular to intracellular volume, and $\delta'$ is the metabolite degradation rate. We now show that by rescaling time, we can eliminate $\alpha$. We first introduce our dimensionless time variable $t' = tu$, where $u$ is a parameter with units of time. Substituting into the above equations yields:

$$\frac{dm_i^{\mathrm{int}}}{dt} = \kappa' u E_i - \alpha u \mathrm{Min}_j(m_j^{\mathrm{int}}) + P' u \cdot (m_i^{\mathrm{ext}}/r_V - m_i^{\mathrm{int}}) - \delta' u m_i^{\mathrm{int}}, \tag{10}$$

$$\frac{dm_i^{\mathrm{ext}}}{dt'} = -P' u \cdot (m_i^{\mathrm{ext}}/r_V - m_i^{\mathrm{int}}) - \delta' u m_i^{\mathrm{ext}}. \tag{11}$$

By defining $u = 1/\alpha$, $\kappa = \kappa'/\alpha$, $P = P'/\alpha$, and $\delta = \delta'/\alpha$, we recover *Equations 1; 2*.

We present transport in our model as passive exchange with the environment, but these equations describe a broader range of transport phenomenon. In particular, our expressions for metabolite exchange are mathematically equivalent to active transport in the linear regime. To demonstrate this, suppose that import and export are mediated by two separate enzymes with different kinetics. In the linear regime, the transport expression will be $\eta_1 m_i^{\mathrm{ext}}/r_V - \eta_2 m_i^{\mathrm{int}}$ where the $\eta$ are constants defining transporter performance (for Michaelis-Menten kinetics, each $\eta$ would be the ratio of the maximum rate and the half-saturation constant of the corresponding transporter). Rearranging terms, one obtains $\eta_2 \left( \frac{\eta_1}{\eta_2 r_V} m_i^{\mathrm{ext}} - m_i^{\mathrm{int}} \right)$ such that there is an effective permeability $\eta_2$ and effective volume ratio $\frac{\eta_2 r_V}{\eta_1}$.

## Appendix 2

## Hybrid numerical methods

In our two-metabolite model, the enzymes are produced in intermittent bursts which must be modeled stochastically, while other reactions in the system (such as metabolite consumption) involve a large enough number of molecules or occur frequently enough that they can be treated deterministically. In order to efficiently simulate this system, we use a hybrid stochastic-deterministic algorithm that can be viewed as a generalization of the Gillespie algorithm. We briefly present the rationale for the algorithm here, with a full description available in (*Griffith et al., 2006*).

We begin with the traditional Gillespie algorithm. In this algorithm, the time between events is an exponential random variable with mean $1/r_{tot}$, where $r_{tot}$ is the sum of all reaction rates in the system. This formulation assumes that $r_{tot}$ is constant between stochastic reactions, something that will not be true in a system that also includes deterministic reactions. To account for time-varying reaction rates, we first recognize that the Gillespie event condition can be rewritten as:

$$\int_t^{t+\tau} r_{tot}dt = r_{tot}\tau \sim \text{Exponential}(1), \tag{12}$$

where $\tau$ is the stochastic time before the next reaction. This rewritten condition makes intuitive sense: if one regards the integral on the left-hand side as a kind of cumulative probability, this states that for every one unit of cumulative probability, on average one event occurs. From *Equation 12* we can rigorously construct a simulation algorithm for systems with time-varying reaction rates. We first sample a random number $x_1 \sim \text{Exponential}(1)$. We then integrate the deterministic dynamics until we find a $\tau$ such that:

$$\int_t^{t+\tau} r_{tot}(t)dt = x_1. \tag{13}$$

In our method, the integration is done by Euler's method. We then sample another random number $x_2 \sim \text{Uniform}(0, 1)$ and use this to determine which reaction occurs, as in the traditional Gillespie method. The reaction event then occurs at time $t + \tau$, and we repeat the above procedure until the simulation reaches a termination condition. In our simulation, the only stochastic events are the enzyme production bursts, while all other reactions are modeled deterministically. For calculation of CVs and growth rates in *Figures 1 and 2*, simulations were run for $4 \times 10^5$ time units (which corresponds to 5000 time units when normalized to the maximum steady-state growth rate in *Figure 1*), with statistics computed from the last $3 \times 10^5$ time units. This time is more than sufficient for the system to forget its initial conditions, as can be seen in *Appendix 9—figure 7*. Simulations are run with initial conditions $m^{int}(t = 0) = m^{ext}(t = 0) = \frac{\kappa\gamma}{2(1+\delta+P)}$ and $E(t = 0) = \gamma/2$. This metabolite concentration is the mean steady-state intracellular metabolite concentration assuming an infinite extracellular volume, and the enzyme concentration is the mean steady-state enzyme concentration. For *Figure 1E, F*, 140 replicate simulations were run per point. For *Figure 2B, C*, 40 replicate simulations were run for most points, with 200 replicates run for certain simulations with large burst size and low permeability.

## Appendix 3

### Langevin description

To better understand the influence of community size on metabolite noise, we consider a simplified version of the two-metabolite model. In this model, we only track the concentration of a single constitutively produced enzyme and its corresponding metabolite. The Langevin equations governing the dynamics are:

$$\frac{dE}{dt} = n\Gamma - \mu_E E + \xi_E(t), \tag{14}$$

$$\frac{dm}{dt} = \kappa E - \mu_m m + \xi_m(t). \tag{15}$$

where the Langevin noise terms have zero noise $\langle \xi_E(t) \rangle = \langle \xi_m(t) \rangle = 0$ and are delta-correlated such that $\xi_i(t_1), \xi_j(t_2) \rangle = \Omega_{ij} \delta(t_1 - t_2)$, where $\delta(\cdot)$ is the Dirac delta function. These equations have mean steady-state fixed point at $E^* = \frac{n\Gamma}{\mu_E}$ and $m^* = \frac{\kappa n\Gamma}{\mu_E \mu_m}$. To analytically compute the CV of the enzyme and metabolite, we employ the method of Swain (**Swain, 2004**). This method relies on linearizing the system about steady state and assumes that fluctuations about the steady state are sufficiently small so as not to drive the system out of the linear regime around the fixed point. Since our system is linear to begin with, this method will actually provide us with exact expressions of the moments.

We begin by computing the elements of $\Omega$. The squared deviation of $\xi_E(t)$ will obey:

$$\langle \xi_E^2(t)\delta t^2 \rangle = \left( \beta^2 \left( \frac{n\Gamma}{\beta} \right) + (-1)^2 \mu_E E^* \right) \delta t, \tag{16}$$

$$\Omega_{EE} = \langle \xi_E^2(t)\delta t \rangle = n\Gamma \left( \beta + 1 \right). \tag{17}$$

Similarly, we can compute the value of $\Omega_{mm}$:

$$\Omega_{mm} = \langle \xi_m^2(t)\delta t \rangle = \frac{2\kappa\Gamma n}{\mu_E}. \tag{18}$$

The off-diagonal entries of $\Omega$ will be zero, as the two equations share no common reactions.

Next, we represent our deterministic dynamics as a matrix equation by computing the Jacobian about the fixed point

$$A = \begin{pmatrix} -\mu_E & 0 \\ \kappa & -\mu_m \end{pmatrix}. \tag{19}$$

This Jacobian has eigenvalues $\lambda_1 = -\mu_E$ and $\lambda_2 = -\mu_m$ with a matrix of column eigenvectors

$$B = \begin{pmatrix} \frac{\mu_m - \mu_E}{\kappa} & 0 \\ 1 & 1 \end{pmatrix}. \tag{20}$$

Variances and covariances of the state variables can then be computed using the following expression:

$$\langle [X_i - \langle X_i \rangle][X_j - \langle X_j \rangle] \rangle = -\sum_{p,q,r,s} B_{ip} B_{jr} \left( \frac{\Omega_{qs}}{\lambda_p + \lambda_r} \right) B_{pq}^{-1} B_{rs}^{-1}, \tag{21}$$

where the $X_i$ are the state variables.

Substituting in the values of $B$, $\Omega$, and $\lambda$ yields

$$\text{Var}[E] = \frac{\Gamma n(\beta+1)}{2\mu_E}, \tag{22}$$

$$\text{Var}[m] = \frac{\kappa\Gamma n(\beta\kappa + 2\mu_m + 2\mu_E + \kappa)}{2\mu_E \mu_m(\mu_E + \mu_m)}. \tag{23}$$

From these expressions for the variance, we compute the CVs as

$$\text{CV}_E = \frac{\sqrt{\text{Var}[E]}}{\text{Mean}[E]} = \frac{\sqrt{\text{Var}[E]}}{E^*} = \sqrt{(\beta+1)\left(\frac{\mu_E}{2\Gamma n}\right)},$$

(24)

$$\text{CV}_m = \frac{\sqrt{\text{Var}[m]}}{\text{Mean}[m]} = \frac{\sqrt{\text{Var}[m]}}{m^*} = \sqrt{\frac{\mu_E \mu_m (\kappa\beta + 2\mu_E + 2\mu_m + \kappa)}{2\kappa\Gamma n(\mu_E + \mu_m)}}.$$

(25)

## Appendix 4

### Generalized NAC framework

To compute distributions of growth rates from metabolite distributions, we first compute the growth distribution's CDF in accord with *Equation 7* and then apply numerical differentiation to yield the PDF. To obtain the expected value, we numerically integrate according to the expression $\mathbb{E}[X] = \int_{-\infty}^{\infty} xp(x)dx$ where $p(x)$ is the probability density of value $x$. As a reference distribution, we used the median gamma distribution of *E. coli* essential proteins observed in (*Taniguchi et al., 2010*) ($k = 6.4$ and $\theta = 5.2$, taken from Table S3 of Taniguchi et al.). For distributions with different CVs, we maintain the same mean as the median distribution of *E. coli* essential proteins.

The case of correlated metabolite levels was analyzed by simulation. To generate correlated distributions, we first generate 50,000 samples for each metabolite from a multivariate normal distribution with the desired correlation structure. The MATLAB function mvnrnd is used for this purpose. We then use the method of copulas to generate samples from correlated gamma distributions (*Nelsen, 2007*). This method first involves transforming the normally distributed samples into uniformly distributed samples using the inverse CDF of the normal distribution. We then transform these uniformly distributed samples into gamma distributed samples using the appropriate gamma CDF. Note that this method is only guaranteed to exactly preserve the rank correlation, but we find that the linear correlation is also very well preserved.

## Appendix 5

### Detailed conditions for noise-driven growth reduction

Here, we present a more general set of conditions for metabolite noise to reduce growth. In the main text, we invoked Jensen's inequality to argue that metabolite noise will reduce growth if a growth function is concave. This statement is true, but it is not a complete description: there are many non-concave functions that can result in reduced growth in the presence of noise. In addition, as mentioned in the discussion, growth sensitivity to noise is not only a property of the growth function, but also of the metabolite distribution. The most general requirement is that, given a growth function $g(X)$ and a metabolite distribution $f(x)$, the 'Jensen gap' $\mathbb{E}[g(\mathbf{X})] - g(\mathbb{E}[\mathbf{X}])$ must be negative.

To show that noise-dependent growth reduction can occur with non-concave functions, we consider two toy examples. First, consider a growth function that is largely concave, but is convex at low nutrient levels (this convexity could reflect a minimum nutrient level required for growth). We show such a function as the red curve of *Appendix 9—figure 8*. For metabolites uniformly distributed across the entire plotted growth function, the Jensen gap will still be negative despite the initial region of convexity in the growth function. Now consider a more extreme case: a growth function that is everywhere oscillatory, rapidly switching between concave and convex, but has a mean behavior matching a concave function. We show such a function as the black curve in *Appendix 9—figure 8*. As in the previous case, this function has a negative Jensen gap for uniformly distributed metabolites.

In order to determine whether noise will reduce growth rate for a particular function and distribution pair, one must estimate the Jensen gap. Analytically, this can be done using "sharpened" versions of Jensen's inequality. These expressions do not provide exact bounds, but provide more accurate results than Jensen's inequality. For example, Theorem 1 of (*Liao and Berg, 2018*) provides an simple inequality that more tightly bounds the Jensen gap.

## Appendix 6

### Chemostat NAC experiment theory

Here, we present the theory for an experiment in which the effects of NAC are measured in a chemostat. First, we begin with the equations governing the chemostat dynamics:

$$\frac{d\rho}{dt} = \mu\rho\left(\frac{S}{K+S}\right) - \delta\rho, \tag{26}$$

$$\frac{dS}{dt} = \Gamma\delta - \mu\rho\left(\frac{S}{K+S}\right) - \delta S \tag{27}$$

where $\rho$ is the biomass concentration, μ is the maximum growth rate, $S$ is the nutrient concentration, $K$ is the half-saturation constant, $\delta$ is the dilution rate, and $\Gamma$ is the inlet nutrient concentration. Note that we are measuring biomass and nutrients in the same units. The steady-state nutrient value will be:

$$S^* = \frac{K\delta}{\mu-\delta}. \tag{28}$$

The steady-state biomass will therefore be:

$$\rho^* = \Gamma - \frac{K\delta}{\mu-\delta}. \tag{29}$$

Thus, if we model NAC as an increase in $\mu$, a higher than expected dilution rate will be required to maintain a given high cell density. To test this, one can estimate $\mu$ and $K$ at low cell densities and compute the predicted density as a function of dilution rate using *Equation 29*. Then, the actual steady-state densities can be measured for a range of dilution rates. If NAC is occurring, the predictions will match the experimental data at high dilution rates, while there will be a significant discrepancy between predicted and observed densities at low dilution rates.

## Appendix 7

### NAC metabolites within a biofilm

In this section, we estimate how far leaked NAC metabolites can travel in a biofilm. Consider a biofilm with cylindrical symmetry and radial coordinate $\hat{r}$. In the center of the biofilm is a sector of radius $\hat{a}$ composed of cells engaging in NAC, while the outer shell of cells are cheaters that only uptake metabolite and do not engage in NAC. We model metabolite leakage in the NAC sector as a constant metabolite production rate $\xi$. In both sectors, metabolites are imported at a rate of $\mu m(\hat{r})$, where $\mu$ is the uptake rate and $m(\hat{r})$ is the local metabolite concentration. Note that for simplicity we are assuming that the metabolite concentrations are below the saturation point of the uptake transporters. Within the NAC sector ($\hat{r} < \hat{a}$), the steady-state concentration of $m$ will obey

$$0 = D \left( \frac{1}{\hat{r}} \frac{\partial m}{\partial \hat{r}} + \frac{\partial^2 m}{\partial \hat{r}^2} \right) - \mu m + \xi, \tag{30}$$

where $D$ is the diffusion coefficient of the metabolites within the biofilm. We assume a no-flux boundary at $\hat{r} = 0$

$$\left. \frac{\partial m}{\partial \hat{r}} \right|_0 = 0, \tag{31}$$

and an arbitrary flux at $\hat{r} = \hat{a}$ such that

$$-D \left. \frac{\partial m}{\partial \hat{r}} \right|_{\hat{a}} = J_{\text{out}}. \tag{32}$$

Outside of the NAC sector ($\hat{r}$), the steady-state concentration of $m$ obeys:

$$0 = D \left( \frac{1}{\hat{r}} \frac{\partial m}{\partial \hat{r}} + \frac{\partial^2 m}{\partial \hat{r}^2} \right) - \mu m. \tag{33}$$

We assume a zero metabolite boundary condition at infinity

$$\lim_{\hat{r} \to \infty} m(\hat{r}) = 0, \tag{34}$$

and the same flux boundary condition at $\hat{r} = \hat{a}$ as with *Equation 30*. Once we solve the boundary problems in the two sectors, we will solve for $J_{\text{out}}$ by requiring continuity of $m$ at $\hat{r} = \hat{a}$.

We introduce a non-dimensionalized radial coordinate $r\nu = \hat{r}$ where $\nu = \sqrt{\frac{D}{\mu}}$. With this non-dimensionalization, we can rearrange *Equation 33* into an inhomogeneous modified Bessel equation of zeroth order:

$$r \frac{\partial m}{\partial r} + r^2 \frac{\partial^2 m}{\partial r^2} - (r^2 + 0^2)m = -r^2 \frac{\xi}{\mu}. \tag{35}$$

Similarly, *Equation 30* reduces to a homogeneous modified Bessel equation of zeroth order:

$$r \frac{\partial m}{\partial r} + r^2 \frac{\partial^2 m}{\partial r^2} - (r^2 + 0^2)m = 0. \tag{36}$$

With this non-dimensionalization, the boundary condition at $\hat{r} = \hat{a}$ becomes:

$$-\left. \frac{\partial m}{\partial r} \right|_a = \frac{J_{\text{out}}}{\sqrt{\mu D}}, \tag{37}$$

where $a = \hat{a} \sqrt{\frac{\mu}{D}}$.

We begin by solving the boundary value problem outside of the NAC sector. The general solution of *Equation 36* is a linear combination of modified Bessel functions

$$m(r) = c_1 I_0(r) + c_2 K_0(r), \tag{38}$$

where $I_0$ and $K_0$ are the zeroth order modified Bessel functions of the first and second kind, respectively. Applying boundary conditions from *Equations 37 and 34* yields:

$$m(r) = \left( \frac{J_{\text{out}}}{\sqrt{\mu D}} \right) \left( \frac{K_0(r)}{K_1(a)} \right) \qquad r > a. \tag{39}$$

This solution is consistent with intuition: modified Bessel functions of the second kind are monotonic decreasing and the solution reduces to $m(r) = 0$ for $J_{\text{out}} = 0$. Similarly, we can obtain the solution to the boundary-value problem within the NAC sector:

$$m(r) = \frac{\xi}{\mu} - \left( \frac{J_{\text{out}}}{\sqrt{\mu D}} \right) \left( \frac{I_0(r)}{I_1(a)} \right) \qquad r < a. \tag{40}$$

This solution is also consistent with intuition: modified Bessel functions of the first kind are monotonic increasing and the solution reduces to $m(r) = \xi/\mu$ if $J_{\text{out}} = 0$. Now that we have the solutions for each sector, we can solve for the value of $J_{\text{out}}$ by requiring $m$ to be continuous between both solutions at $r = a$:

$$J_{\text{out}} = \frac{\xi}{\sqrt{\frac{\mu}{D}} \left( \frac{K_0(a)}{K_1(a)} + \frac{I_0(a)}{I_1(a)} \right)}. \tag{41}$$

To parameterize this model, we use values measured from a *Vibrio cholerae* biofilm in which N-Acetylglucosamine is a public good that is being produced and consumed (**Drescher et al., 2014**). In this system, $D = 500\mu\text{m}^2/\text{s}$ and $\mu = 3.4 \times 10^3 \text{ s}^{-1}$. The value of $\xi$ cannot be easily estimated, but as long as metabolite uptake is linear, $\xi$ will only rescale the solution and not alter the boundary size. We take a high estimate of $\xi = 2.5 \times 10^6$ molecules/$\mu\text{m}^3$s, approximately equivalent to *E. coli* leaking its entire aspartic acid pool each second. We plot metabolite profiles for two different values of $\hat{a}$ in *Appendix 9—figure 9*. As expected, the length of the boundary region in which the NAC metabolites are available to cheaters is approximately constant. This occurs because the decay length scale of the modified Bessel function of the second kind only depends on $\mu$ and $D$, not $\hat{a}$. Therefore, for large NAC sectors the number of cells in the transition region is small compared to the number of cells in the NAC sector.

## Appendix 8

### Models without metabolite degradation

To explore the impact of metabolite degradation on our results, we now analyze a version of the metabolite model without metabolite degradation (i.e. with $\delta = 0$). We first examine the behavior of individual cells by repeating the analyses of *Figure 1EF*. These results can be found in *Appendix 9—figure 10A,B*. How does permeability influence growth in a system where metabolites do not degrade? In *Appendix 9—figure 10A*, we plot the growth rate of single cells as a function of permeability at different values of burst size. Curiously, the cell's growth is poor at low permeability, improving when permeability is moderate. For $\beta = 10$ and $\beta = 20$, growth decreases again at very high permeability. This is in stark contrast to the case of single cells with metabolite degradation, in which increased permeability simply lowers growth. We found that this non-monotonic behavior arises due to a 'Paradox of Plenty' in which increased intracellular metabolite build-up can make a cell more likely to die via sudden metabolite 'crashes'. We show the metabolite and enzyme timecourses of a cell undergoing this crash behavior in *Appendix 9—figure 10C,D*. At around $t = 15$, the cell develops an excess of the blue metabolite with a deficit of the pink metabolite, limiting its growth. In the time it takes the organism to produce a new burst of the pink enzyme, it continually builds up its pool of the blue metabolite, as there is no degradation term to limit the pool's size. Eventually, at around $t = 35$, the cell produces a burst of the pink enzyme, generating more pink metabolite and allowing a resumption of steady growth. Since there is such a large build-up of the blue metabolite, it takes the cell a long time to deplete this metabolite pool. In this time, the cell produces many bursts of the pink enzyme, almost entirely replacing the cell's enzyme pool and leaving the cell with almost no blue enzyme. At around $t = 55$, the blue metabolite crashes, rapidly becoming the new limiting metabolite. At this point, the cell is left in a difficult position, suddenly having almost none of the now limiting blue metabolite and having an enzyme pool entirely dedicated to production of the pink metabolite. Growth rapidly ceases, as the remaining blue metabolite pool is insufficient to produce a corrective enzyme burst. Even if the cell had survived, it would once again be back to having a severe imbalance, eventually leading to another crash. Such crashes can be averted if the cell is able to transfer a portion of its metabolites to an extracellular space, reducing the magnitude of intracellular metabolite build-up. For this reason, increasing permeability initially promotes cell growth, as seen in *Appendix 9—figure 10A*. However, if permeability becomes too high, the intracellular and extracellular metabolites are always equilibrated and metabolites can flow rapidly back into the cell. In this case, the cell is less able to avert metabolite build-up, leading again to decreased growth for cells with moderate burst sizes.

In *Appendix 9—figure 10B*, we show the growth rate of isolated cells as a function of burst size. As expected, the growth rate decreases with burst size. The non-monotonic relationship between growth and permeability also manifests in these results, with the $P = 1$ case having the highest growth. Next, we explore the impact of growth on multicell communities. In *Appendix 9—figure 10E*, we show the growth rate of ten-cell communities as a function of permeability. For lower burst sizes, the effect of NAC persists, with growth rate increasing with permeability. The lack of degradation actually allows NAC to be beneficial for cells with $\beta = 10$, where previously they did not benefit from NAC because of losses due to degradation. Interestingly, NAC is no longer sufficient to rescue growth for burst size $\beta = 100$. This is due to the aforementioned Paradox of Plenty, which makes it even more difficult for these high-burst-size cells to regulate their metabolism. In *Appendix 9—figure 10F* we show the corresponding CVs as a function of permeability. Notably, the trends are non-monotonic, with increased permeability initially reducing noise, but eventually leading to increased noise. The initial decrease in noise arises from the NAC mechanism of noise reduction discussed in the main text. The eventual increase in noise with high permeability is due to the metabolite build-ups discussed above. Once the cells are strongly connected and their metabolite pools are in sync, they are more prone to large metabolite build-ups. However, while these build-ups increase noise, the linking of cells allows for increased overall metabolite production, leading to an increase in mean metabolite levels that outweighs the increase in variance.

## Appendix 9

### Supplemental figures

In this section, we present supplemental figures that support the main text. Each figure's caption contains all pertinent information.

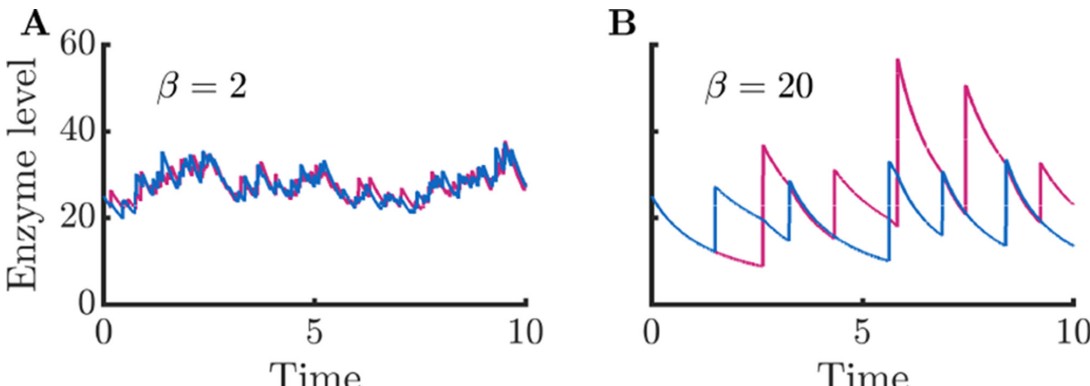

**Appendix 9—figure 1.** Enzyme timecourses corresponding to *Figure 1CD*. (**A**) Enzyme timecourse of a cell that produces enzymes in small bursts, same parameters as in *Figure 1C*. (**B**) Enzyme timecourse of a cell that produces enzymes in large bursts, same parameters as in *Figure 1D*. Note that the enzyme levels are substantially correlated with each other ($r = 0.89$ for $\beta = 2$ and $r = 0.53$ for $\beta = 20$). This correlation between enzyme levels also results in correlation between metabolite levels ($r = 0.8$ for $\beta = 2$ and $r = 0.25$ for $\beta = 20$).

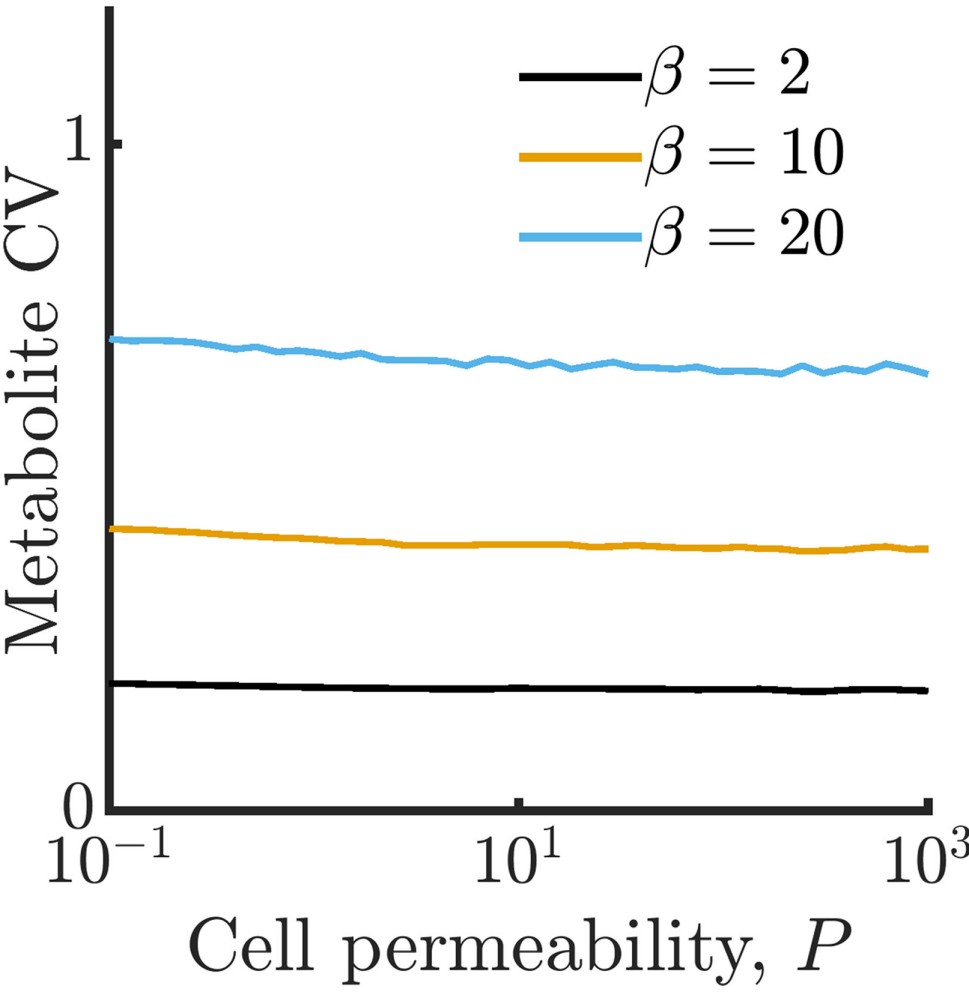

**Appendix 9—figure 2.** Metabolite CV corresponding to simulations in *Figure 1E*. All simulation parameters identical to those in *Figure 1E*. Data corresponding to $\beta = 100$ not shown as metabolite CV cannot be meaningfully estimated for cells with arrested metabolism.

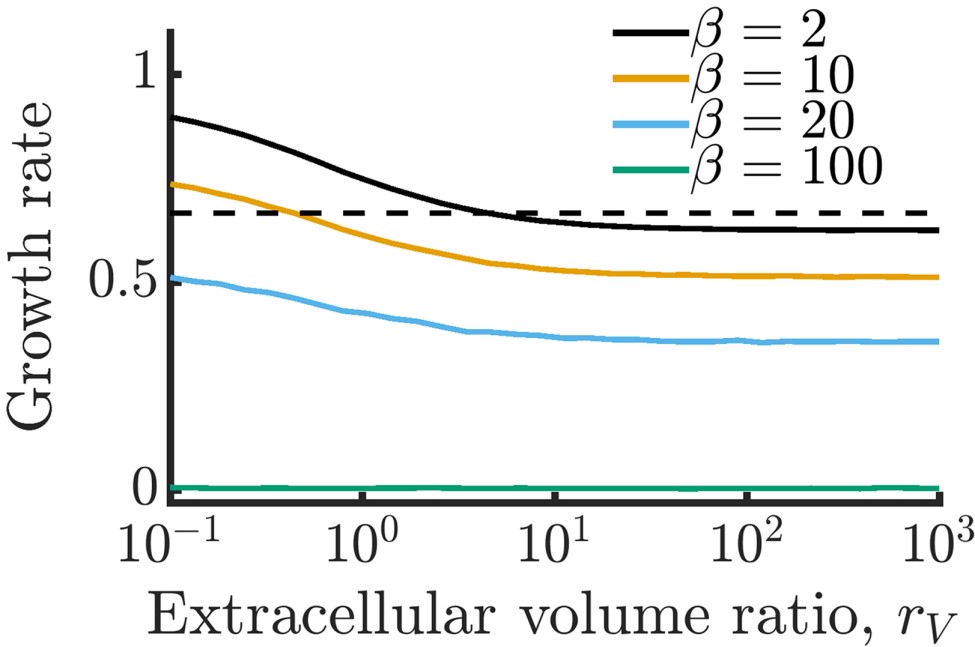

**Appendix 9—figure 3.** Impact of extracellular volume on growth rate of isolated cells. As extracellular volume is increased, cells are increasingly less likely to recover their leaked metabolites, leading to decreased growth. This effect plateaus once the extracellular volume becomes effectively infinite such that cells never recover their leaked metabolites. The dashed black line represents the maximum growth rate of cells at infinite volume, calculated by treating the permeability as an additional intracellular metabolite degradation term (i.e. $g = \frac{\kappa\gamma g^*}{2(1+\delta+P)}$). Parameters used are the same as in 1E with a permeability of $P = 1$. Values presented are averaged from 100 replicate simulations.

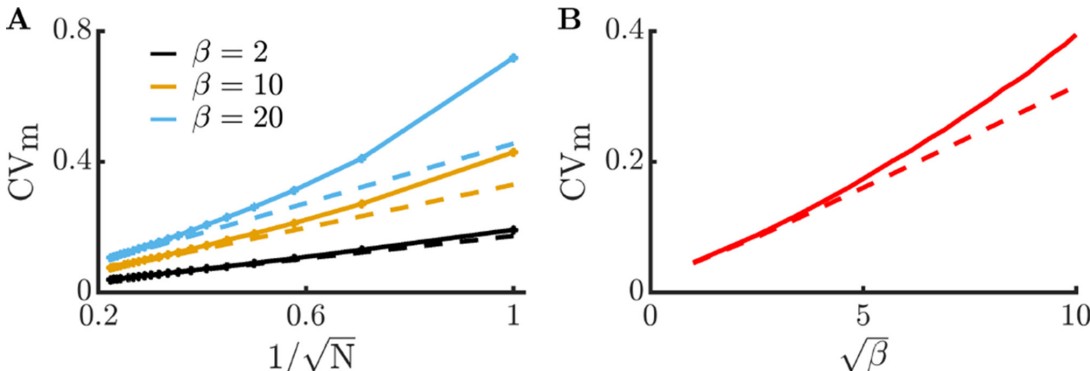

**Appendix 9—figure 4.** Comparison of Langevin equation predictions to full metabolism model. The full model is simulated with the same parameters as in *Figure 1EF*, except with $V = 0$ and $P = 100$. We apply the following substitutions to the Langevin model in order to directly compare it to the full model: $\Gamma = \gamma g_{max}$, $\mu_E = g_{max}$, and $\mu_m = \delta + 1$, where $g_{max} = \frac{\kappa\gamma g^*}{2(1+\delta)}$. (**A**) Comparison of metabolite CV in the full model (solid curves) and the Langevin model (dashed curves) as a function of community size at different burst sizes. (**B**) Comparison of metabolite CV in the full model (solid curve) and the Langevin model (dashed curve) as a function of burst size. As expected, the agreement between the full model and the Langevin model is best at high $N$ and low $\beta$ - conditions that lead to low levels of noise.

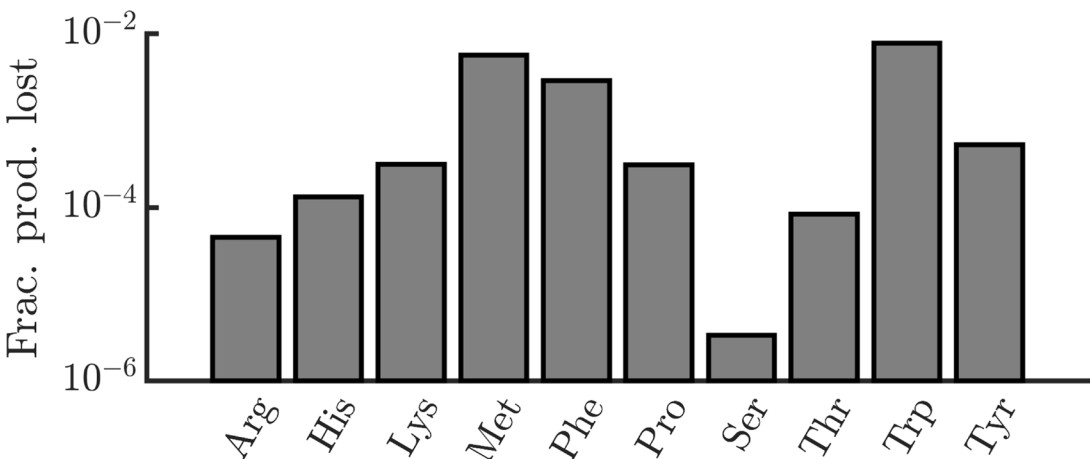

**Appendix 9—figure 5.** Estimated fractional production losses of amino acids in *E. coli* due to leakage. *E. coli* due to leakage. In order to determine the significance of amino-acid leakage in *E. coli*, we estimated the fraction of *E. coli*'s amino-acid production that is lost to leakage. Formally, we define the fraction of production lost as the ratio of the number of amino acids lost via leakage over the period of one division to the number of amino acids required to produce a daughter cell. For each amino acid, we require three experimentally measured quantities for this calculation: (1) the rate of leakage through the cell membrane, (2) the intracellular concentration of the amino acid, and (3) the number of amino acids required to produce a daughter cell. For leakage rates, we use data from artificial liposomes (**Chakrabarti and Deamer, 1992**). In cases where multiple pH conditions were tested, we used data measured at pH 7 (though leakage rates did not vary substantially with pH). This study measured data for only a limited set of amino acids. For other amino acids, we estimated their leakage rates using a linear regression of leakage rate versus log octanol/water partition coefficient ($r^2 = 0.96$). Leakage rates were also adjusted for the differing size of the liposomes and typical *E. coli* cells, assuming a liposome radius of 100 nm and an *E. coli* radius of 400 nm (**Grossman et al., 1982**). Intracellular concentrations were taken from (**Bennett et al., 2009**) and per-cell pool sizes were calculated assuming a cell volume of $1.8 \times 10^{-15} L$ (**Outten and O'Halloran, 2001**). A cell's amino-acid production was assumed to be the number of amino acids required to produce a daughter cell, taken from (**Mee et al., 2014**). With all of these experimental values, the fraction of production lost is $f = \frac{kN_I\tau}{N_{tot}}$ where $k$ is the leakage rate, $N_I$ is the intracellular molecule count, $\tau$ is the doubling time (assumed to be 24 minutes), and $N_{tot}$ is the number of amino acids required to produce a daughter cell.

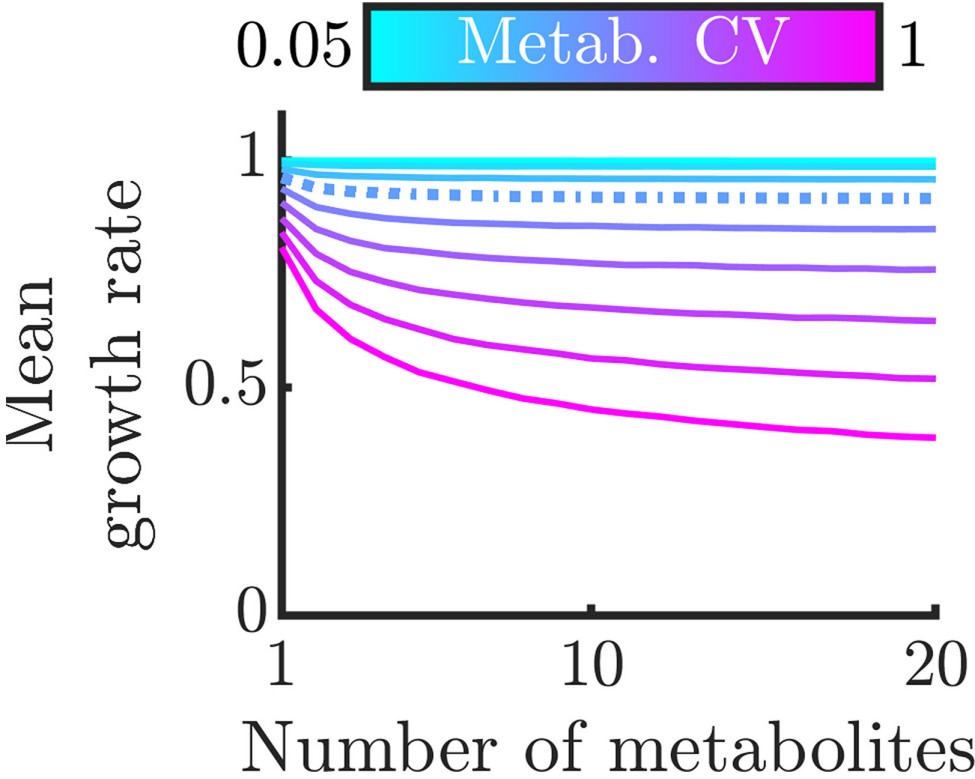

**Appendix 9—figure 6.** Version of **Figure 3C** with an alternative growth function. To demonstrate that our findings are not limited to Liebig's law of the minimum, we repeat the analysis in **Figure 3C** with an alternative growth function from (**Goyal and Wingreen, 2007**). The function is $g = \frac{g_{max}}{\frac{1}{N}\sum_{i=1}^{N}\frac{m_i+m_i^*}{m_i}}$, where $g_{max}$ is the maximum growth rate, $N$ is the total number of unique metabolites, and $m_i^*$ is the half-substrate constant of each metabolite. In this analysis we assume $g_{max} = 1$ and $m_i^* = \mathbb{E}[m_i]$.

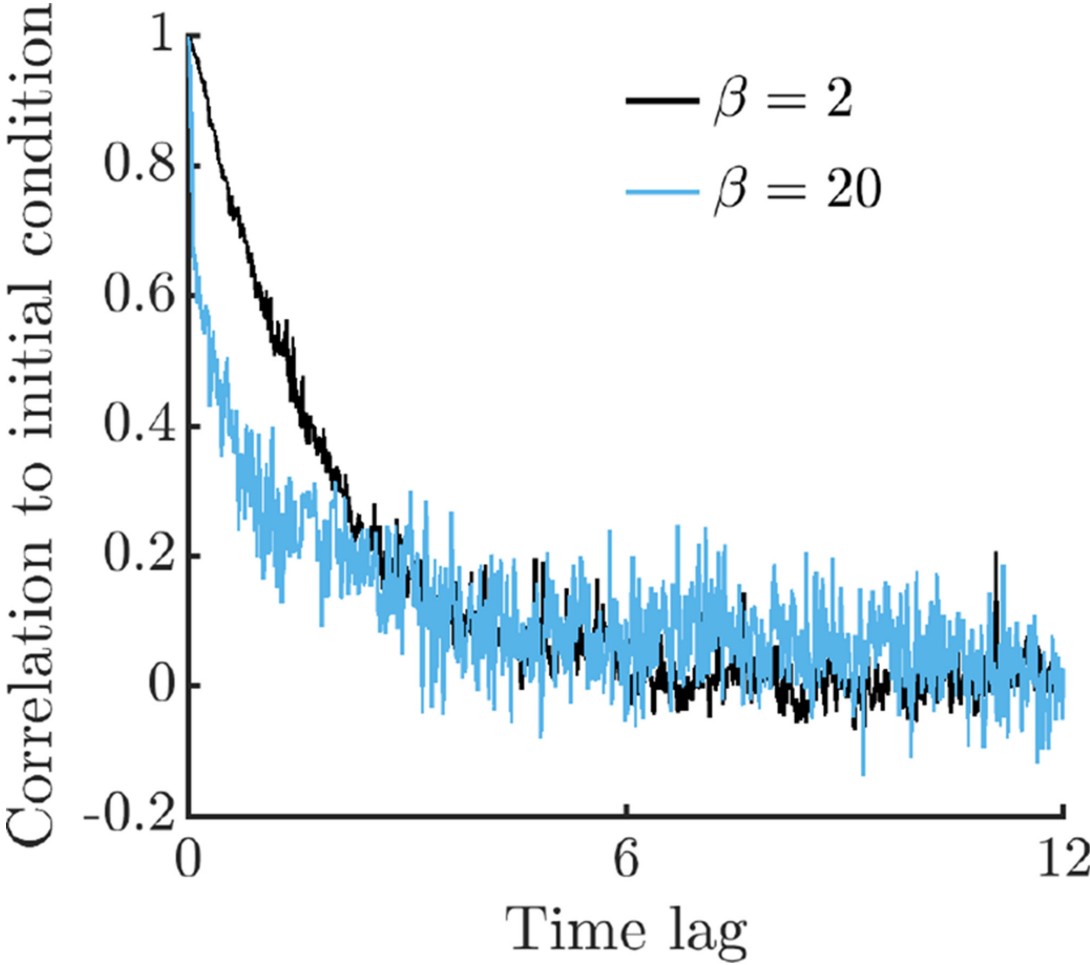

**Appendix 9—figure 7.** Correlation between initial conditions and later times in the full metabolism model. To compute correlations to the initial condition, we perform 2,500 replicate simulations of single-cell metabolism with two different values of burst size $\beta$, each with the same initial conditions except for the value of $E_1$, which is uniformly distributed in the range $[0, \gamma]$. Simulations are run for 12.5 time units, where time is normalized to the maximum steady-state growth rate, as in *Figure 1*. We bin the resulting timeseries with a window size of 0.025 time units, computing Pearson correlations between the initial conditions and all data points within the window. From each bin, we also compute the mean time of datapoints within the bin, which we use as the $x$-axis data in this figure. As can be seen, the system rapidly forgets its initial conditions, nearing zero correlation after a short time lag.

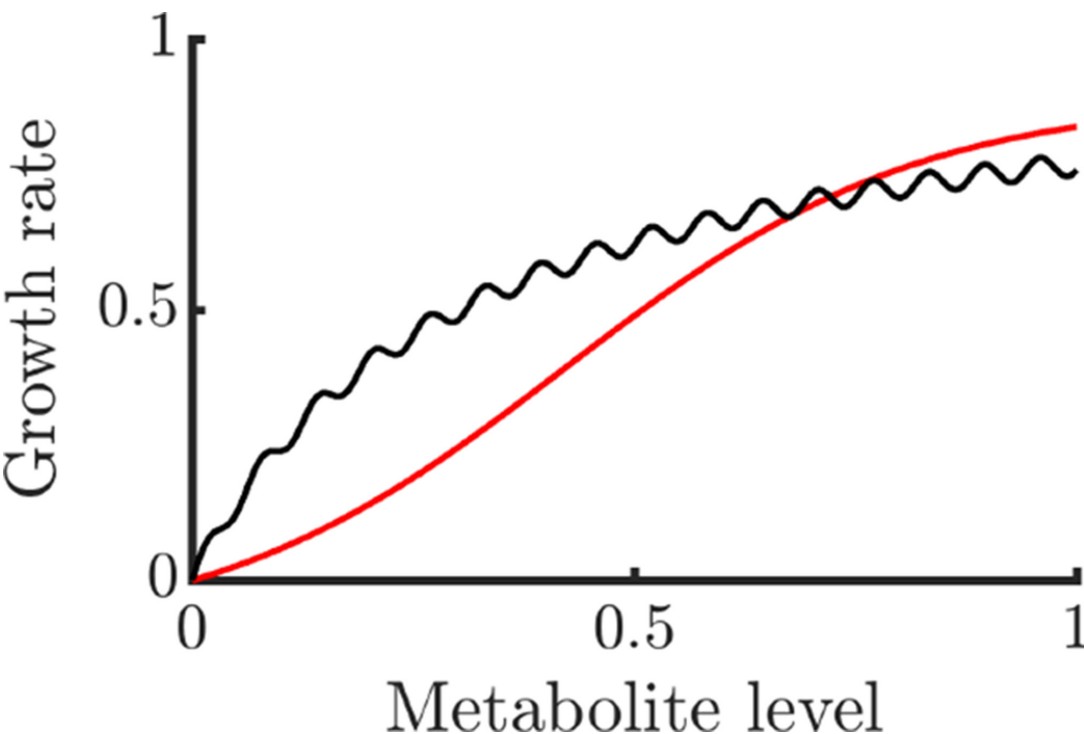

**Appendix 9—figure 8.** Examples of non-globally concave growth functions that still allow NAC to be potentially beneficial. The black curve is $g(x) = x/(0.3 + x) + 0.02\sin(100x)$ and the red curve is $g(x) = 1/(1 + e^{-5x+2.1}) - 1/(1 + e^{2.1})$. For metabolites uniformly distributed in the domain $[0, 1]$, both functions have negative Jensen gaps and are thus negatively impacted by noise.

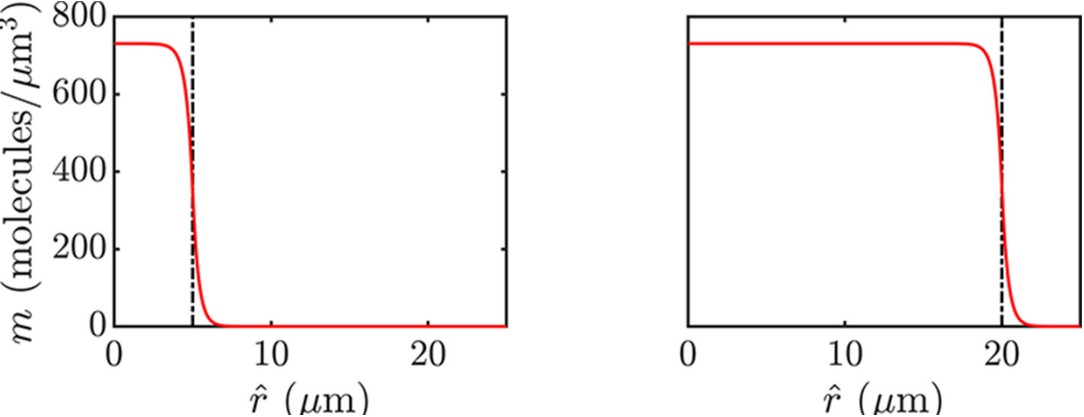

**Appendix 9—figure 9.** Spatial profiles of NAC metabolites within a biofilm. Here, we plot solutions of the biofilm model analyzed and parameterized in Appendix 7. The dashed lines represent the boundary between the NAC and non-NAC sectors of the biofilm, with the NAC sector beginning at $\hat{r} = 0$. Left: Metabolite profiles with a NAC sector of radius $\hat{a} = 5\mu$m. Right: Metabolite profiles with a NAC sector of radius $\hat{a} = 20\mu$m.

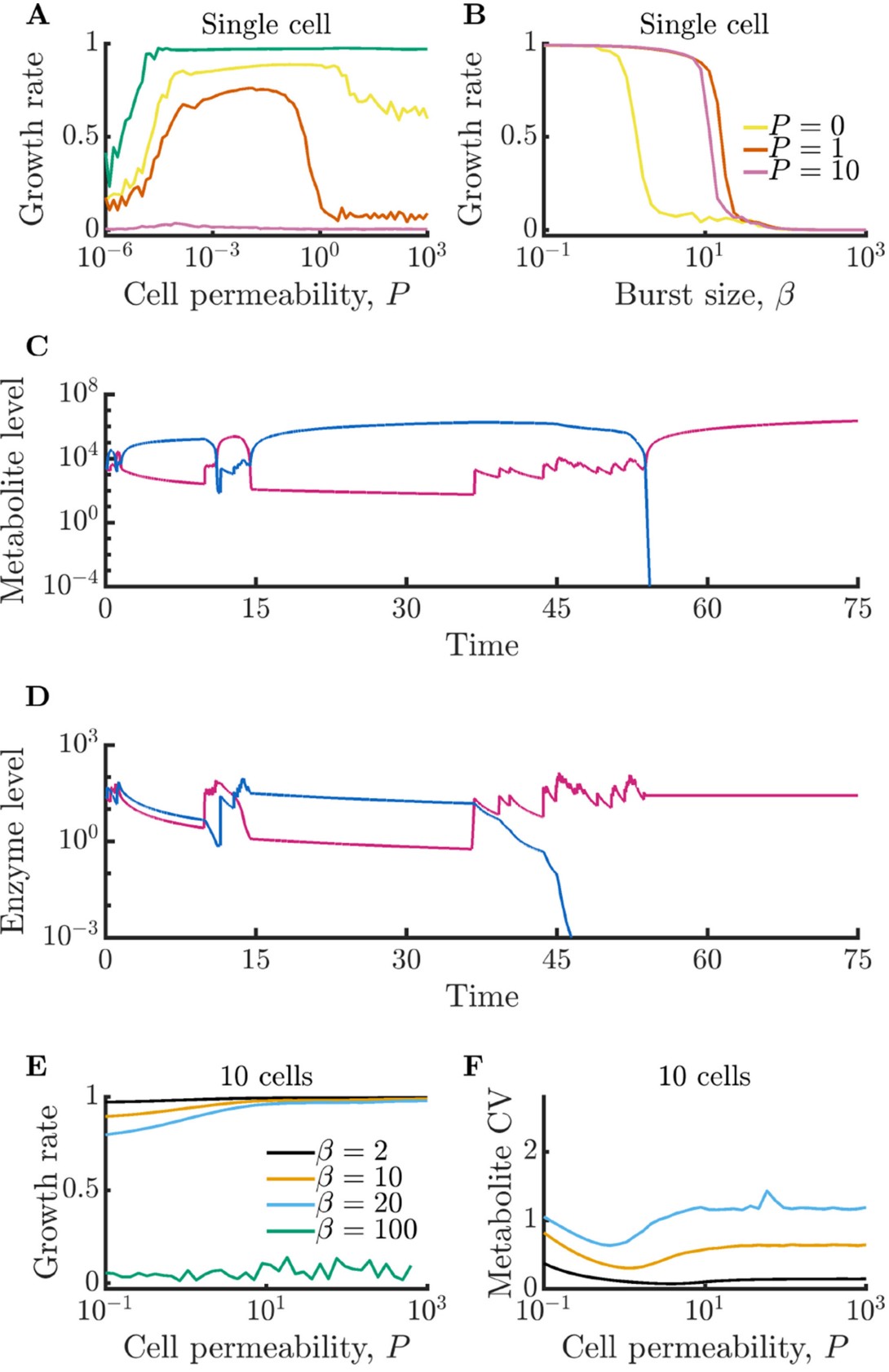

**Appendix 9—figure 10.** Analysis of metabolism model without metabolite degradation. (**A–B**) Versions of *Figure 1EF* with $\delta = 0$. Panel *A* shows an extended range of permeability values to show the non-monotonic

*Appendix 9—figure 10 continued on next page*

*Appendix 9—figure 10 continued*

growth behavior. Data in *A* is computed from 100 replicates and the data from *B* is computed from 200 replicates. All parameters identical to the original figure with the exception of the degradation rate, $\delta$. Simulations are initialized with zero extracellular metabolites. (**C–D**) Timecourses of metabolite and enzyme levels in an isolated cell exhibiting build-up and crash behavior. Parameters the same as in *Figure 1D* with the exception of $P = 0$ and $\delta = 0$. Time normalized by inverse of maximum steady-state growth rate. (**E–F**) Versions of *Figure 2BC* with $\delta = 0$. Data is computed from 40 replicates. All parameters identical to the original figure with the exception of the degradation rate, $\delta$. Simulations are initialized with zero extracellular metabolites.

