## [Editor Report]

In this article, authors propose a novel hypothesis that can help explain why microbes release metabolites. In their NAC (noise-averaging cooperation) hypothesis, within-population cross-feeding can arise due to noisy metabolism in microbes. The authors predict substantial noise-driven growth inefficiencies from single-cell protein abundance data, review evidence for NAC, and propose how to detect NAC in microbial populations.

---

## [Decision Letter]

**Decision letter after peer review:**

Thank you for submitting your article "Noisy metabolism can drive the evolution of microbial cross-feeding" for consideration by *eLife*. Your article has been reviewed by 3 peer reviewers, including Wenying Shou as the Reviewing Editor and Reviewer #1, and the evaluation has been overseen by Naama Barkai as the Senior Editor.

Essential revisions:

Reviewers overall liked this article. Reviewers want to see the evolution of noise-averaging cooperation (NAC), and the evolutionary stability of NAC against cheaters. To help revision, all reviews are attached.

*Reviewer #2 (Recommendations for the authors):*

1. As authors already have discussed, they have shown the optimality of metabolite sharing, not the evolution of cross-feeding. I agree that the authors nicely discussed what scenario could be feasible to evolve cross-feeding from the metabolite sharing. But still, two steps are not explicitly shown: (1) evolutionary stability of sharing metabolites (2) evolution from metabolic sharing to cross-feeding by gene deletion. I do not insist that the authors should examine them, but the current title "Noisy metabolism can drive the evolution of microbial cross-feeding" is misleading. So, it should be revised.

Because of the mismatch between the title and the model, I was confused about what cross-feeding really means. I admit that cross-feeding can happen within a single species, but the model considers identical individuals. It makes me confused that the authors consider metabolite sharing between identical individuals as cross-feeding.

2. Also, the model strongly depends on the bursty behavior of enzyme production, which makes me difficult to find which noise (or stochasticity) is necessary for NAC (or what is a fundamental mechanism of it). It seems that deterministic dynamics with bursty behavior starting from different enzyme concentrations give similar results. If the enzyme production feedback is less frequent, the gap between metabolite concentrations will increase. That induces a lower growth rate. In this case, metabolite sharing will also be optimal.

In the end, I realized that the noise in metabolite concentrations should play a fundamental role not one in the dynamics. But still, I am not sure about the fundamental mechanism, "noise" or "imbalance" of metabolite concentrations. I think the imbalance induces a lower growth rate with a smaller population size leading to a larger CV. Hence, to me, imbalance seems to be the more fundamental reason why metabolite sharing becomes optimal.

3. As I am not an expert in the Black Queen Hypothesis (BQH), it was unclear which question the authors aim to answer in the introduction either the generalization of BQH or the origin of the leakage. At the end of the reading, I found that the authors have asked the latter one. I think this confusion arises because I have no enough background in BQH. Thus, adding an explanation of BQH would help the audience to understand what the main question is.

4. I found that the initial enzyme concentrations are missing in the main text. I guess that the same initial concentrations of all enzymes are used for Figure 1C and 1D, but the authors should mention the initial conditions in the main text. Also, I wonder whether the results are robust under different initial conditions such that the different enzymes have different concentrations.

5. In lines 237~238, the term "curse of dimensionality" is often used in computer science. When the volume of the space increases so fast as the dimensionality increases, the available data becomes sparse. Thus one cannot find the statistical significance. However, in the main text, the authors have used this terminology to emphasize the situation where a large number of metabolites suppress the growth rate. I think the "curse of dimensionality" is inappropriate to be used in this situation.

6. The authors said that the correlated enzyme levels suggest the correlated metabolite levels (lines 248-251). However, in Appendix7 -Figure1, only the correlation between enzyme concentrations is shown without the correlation for the metabolites. It would be nice to verify this assumption by showing the correlation between metabolite concentrations in simulations.

7. As authors explained in lines 253-256, the correlation between metabolites could make their concentrations even but it is not always so. When each concentration has different average values, one can have an outlying low metabolite level even with a positive correlation. It is because the correlation does not tell the metabolite level itself. I think the results in Figure 3D are obtained at the same fixed average metabolite concentrations. If so, please clarify this.

8. In Equation (3) and (4), noise strength is not given in the main text.

9. Typo in line 53: that -> that

*Reviewer #3 (Recommendations for the authors):*

I enjoyed reading the manuscript: the NAC framework is an interesting new take on the evolution of cross-feeding and the manuscript is well written and organized. However, I have some suggestions for improving the presentation of the model.

1. The relation between NAC and BQH should be rephrased in more neutral terms (see public review).

2. It would be important to also mention the economies of scale hypothesis on cross-feeding evolution (see public review). I suggest briefly mentioning it in the intro and discussing how it relates to NAC in more detail in the discussion. Much of this work has been done in the context of amino-acid cross-feeding (see work by Christian Kost) and this would, e.g., be an appropriate place to discuss this.

3. In the absence of an evolutionary model, I think it is essential to phrase any statements about evolutionary dynamics more carefully and make clear to the reader that additional work is needed to confirm these hypotheses (see public review).

4. As mentioned in the public review I think some of the model assumptions have debatable biological rationale. It would be important to justify why these assumptions were made and to discuss how they affect the main conclusions. Additional simulations to explore the sensitivity of the model to these assumptions could be helpful, but are not necessary, provided that the authors can give an intuitive understanding of how these assumptions affect their conclusions.

5. In lines 142-144 and 185-186 the authors talk about the role of extracellular volume/cell density; however, no explanation is given of how these results were derived. I think these statements need to be supported by either a SI figure or by an intuitive explanation of how r_v_ impacts model predictions.

6. In the analytical analysis presented in lines 187-205 the authors make use of various additional simplifying assumptions. Without additional details it is very hard to judge to what extend they affect the presented results. I think some additional discussion on the rational/effect of these assumptions would be highly beneficial to the reader (alternatively one could add a SI figure to e.g. compare the predictions of Equation 5 with data from simulations).

---

## [Author Response]

Reviewer #2 (Recommendations for the authors):1. As authors already have discussed, they have shown the optimality of metabolite sharing, not the evolution of cross-feeding. I agree that the authors nicely discussed what scenario could be feasible to evolve cross-feeding from the metabolite sharing. But still, two steps are not explicitly shown: (1) evolutionary stability of sharing metabolites (2) evolution from metabolic sharing to cross-feeding by gene deletion. I do not insist that the authors should examine them, but the current title "Noisy metabolism can drive the evolution of microbial cross-feeding" is misleading. So, it should be revised.Because of the mismatch between the title and the model, I was confused about what cross-feeding really means. I admit that cross-feeding can happen within a single species, but the model considers identical individuals. It makes me confused that the authors consider metabolite sharing between identical individuals as cross-feeding.

We thank the reviewer for pointing out that the current title is misleading. We have now changed the title to “Noisy metabolism can promote microbial cross-feeding”.

We agree with the reviewer that our definition of cross-feeding was unclear. We do not consider sharing among identical individuals to be cross-feeding. Instead, we consider this to be a form of cooperation that can eventually lead to cross-feeding when one of the individuals loses its capacity to produce one of the metabolites (thus making the individuals non-identical). We have now added clarifying statements to this effect.

2. Also, the model strongly depends on the bursty behavior of enzyme production, which makes me difficult to find which noise (or stochasticity) is necessary for NAC (or what is a fundamental mechanism of it). It seems that deterministic dynamics with bursty behavior starting from different enzyme concentrations give similar results. If the enzyme production feedback is less frequent, the gap between metabolite concentrations will increase. That induces a lower growth rate. In this case, metabolite sharing will also be optimal.In the end, I realized that the noise in metabolite concentrations should play a fundamental role not one in the dynamics. But still, I am not sure about the fundamental mechanism, "noise" or "imbalance" of metabolite concentrations. I think the imbalance induces a lower growth rate with a smaller population size leading to a larger CV. Hence, to me, imbalance seems to be the more fundamental reason why metabolite sharing becomes optimal.

On the necessity of stochasticity: The reviewer is correct that stochasticity is not necessary for metabolite sharing to be optimal. For NAC to be potentially beneficial, there are two requirements: the metabolite levels must fluctuate from the mean, and the growth function must be concave as a function of metabolite levels. In principle, fluctuations originating from deterministic dynamics could also result in a drop in mean growth rate (for example, due to persistent oscillations around the mean). In our analysis of experimental data, we implicitly refer to all variation from the mean as “noise”, but the underlying mathematics are agnostic to the origin of the variation, i.e. the origin of the variation could be stochastic, deterministic, or some combination. We have now added sentences noting this to the relevant section.

On the necessity of imbalance: In our model, the fundamental mechanism underlying the optimality of metabolite sharing is variable metabolite concentrations coupled to a concave growth function. In general, there is no requirement for an “imbalance” per se of metabolite concentrations. Even with a single metabolite, NAC can still be beneficial. In the particular model we analyze, the requirement for imbalance arises from a feature of Liebig’s Law of the Minimum and is actually a proxy for the concavity requirement. Liebig’s Law of the Minimum is concave for only two or more metabolites. In the case of a single metabolite, it reduces to a linear function that is insensitive to noise because it is neither convex nor concave. To better explain the conditions necessary for noise to be detrimental to growth, we have now added an additional appendix “Conditions for noise-driven growth reduction”.

3. As I am not an expert in the Black Queen Hypothesis (BQH), it was unclear which question the authors aim to answer in the introduction either the generalization of BQH or the origin of the leakage. At the end of the reading, I found that the authors have asked the latter one. I think this confusion arises because I have no enough background in BQH. Thus, adding an explanation of BQH would help the audience to understand what the main question is.

We thank the reviewer for pointing out this opportunity to improve the clarity of our introduction. We have now restructured the introduction to better explain the manuscript’s goals and how these relate to the BQH. The evolution of cross-feeding requires the emergence of two types of organisms: the organism secreting the cross-fed compound and the organism consuming the compound. The BQH provides a mechanism for the evolution of the consuming organism, but leaves open the question of how the leaking organism arises. Our goal is to develop a theory complementary to the BQH that explains the origins of the leaking organism, thus filling a gap in the overall theory of cross-feeding.

4. I found that the initial enzyme concentrations are missing in the main text. I guess that the same initial concentrations of all enzymes are used for Figure 1C and 1D, but the authors should mention the initial conditions in the main text. Also, I wonder whether the results are robust under different initial conditions such that the different enzymes have different concentrations.

We thank the reviewer for pointing out this omission. We now report the initial conditions used in our simulations in the “Hybrid numerical methods” appendix. We also now explore the impact of initial conditions on our simulations in a new SI figure (Appendix 9 – figure 7).

In this figure, we compute the mean correlation between enzyme levels and the simulation state at a later time. We find that the system forgets its initial conditions rapidly, on the timescale of ~5 time units (time units normalized to the inverse of the maximum steady-state growth). The simulations reported in the manuscript are run for 5000 of these time units and statistics are computed from only the final ¾ of the growth timecourse. Thus, we are confident initial conditions do not impact our results.

5. In lines 237~238, the term "curse of dimensionality" is often used in computer science. When the volume of the space increases so fast as the dimensionality increases, the available data becomes sparse. Thus one cannot find the statistical significance. However, in the main text, the authors have used this terminology to emphasize the situation where a large number of metabolites suppress the growth rate. I think the "curse of dimensionality" is inappropriate to be used in this situation.

We thank the reviewer for pointing out that our use of this term may be confusing. We have now removed mention of the term.

6. The authors said that the correlated enzyme levels suggest the correlated metabolite levels (lines 248-251). However, in Appendix7 -Figure1, only the correlation between enzyme concentrations is shown without the correlation for the metabolites. It would be nice to verify this assumption by showing the correlation between metabolite concentrations in simulations.

We now report the correlation coefficients between the metabolites in the relevant figure caption.

7. As authors explained in lines 253-256, the correlation between metabolites could make their concentrations even but it is not always so. When each concentration has different average values, one can have an outlying low metabolite level even with a positive correlation. It is because the correlation does not tell the metabolite level itself. I think the results in Figure 3D are obtained at the same fixed average metabolite concentrations. If so, please clarify this.

We thank the reviewer for pointing out this lack of clarity. Figure 3D is indeed generated assuming all metabolites on average are co-limiting. The reviewer is correct that the benefit of correlated metabolites is strongest when the metabolites are on average co-rate limiting. If one metabolite is on average more limiting, even perfect metabolite correlation will not lead to the optimal growth rate. We have now added additional text to the “Generalized model framework” section to clarify these issues.

8. In Equation (3) and (4), noise strength is not given in the main text.

We now report the noise strengths in the main text.

9. typo in line 53: that -> that

We have now fixed this typo.

Reviewer #3 (Recommendations for the authors):I enjoyed reading the manuscript: the NAC framework is an interesting new take on the evolution of cross-feeding and the manuscript is well written and organized. However, I have some suggestions for improving the presentation of the model.1. The relation between NAC and BQH should be rephrased in more neutral terms (see public review).

We thank the reviewer for pointing out this shortcoming in our discussion of the BQH, as it has helped us better put our study in the appropriate context. We agree with the reviewer that the BQH and NAC should in fact be regarded as complementary. We have now revised our introduction to clarify that NAC and BQH are complementary, and we now address the issue of noise being an evolvable trait in the discussion. We also now distinguish the BQH from the assumption of “inevitable” leakiness, as indeed the BQH is agnostic to the source of the leakiness. Drawing from Reviewer #1’s comments, we also have noted another fundamental difference between NAC and the BQH that makes the two complementary: the BQH focuses on the emergence of consumers, while NAC focuses on the emergence of leakers.

2. It would be important to also mention the economies of scale hypothesis on cross-feeding evolution (see public review). I suggest briefly mentioning it in the intro and discussing how it relates to NAC in more detail in the discussion. Much of this work has been done in the context of amino-acid cross-feeding (see work by Christian Kost) and this would, e.g., be an appropriate place to discuss this.

We have now integrated the economies of scale hypothesis into the manuscript. It is now mentioned in the introduction and elaborated on further in the discussion. We found comparisons between the economies of scale hypothesis and NAC particularly useful in our paragraphs on identifying NAC in natural systems. These comparisons allowed us to further sharpen our experimental and observational predictions by outlining cases in which these two hypotheses would be distinguishable.

3. In the absence of an evolutionary model, I think it is essential to phrase any statements about evolutionary dynamics more carefully and make clear to the reader that additional work is needed to confirm these hypotheses (see public review).

We agree. Therefore, we have now rephrased our mentions of evolutionary dynamics to make it clear that the models we analyze are ecological in nature, and by themselves do not allow us to reach evolutionary conclusions. We have restructured the discussion to prioritize discussion of evolutionary dynamics, including appropriate references and connections to the existing literature on the evolution of cooperation.

4. As mentioned in the public review I think some of the model assumptions have debatable biological rationale. It would be important to justify why these assumptions were made and to discuss how they affect the main conclusions. Additional simulations to explore the sensitivity of the model to these assumptions could be helpful, but are not necessary, provided that the authors can give an intuitive understanding of how these assumptions affect their conclusions.

We thank the reviewer for pointing out that these assumptions merit additional discussion. We address each of these two assumptions below:

Growth feedback on enzyme production: This assumption was added to reflect the fact that, on average, all cellular components must increase at the same rate as growth (otherwise cellular component concentrations become progressively imbalanced over time). However, even in the absence of this feedback, the model can still exhibit noise-averaging cooperation. This can be seen in our analysis of the simplified Langevin equation model. The primary difference between the Langevin model and the full metabolism model is the removal of the growth feedback. We have now added an SI figure (Appendix 9 – figure 4) comparing the two models. In large communities and at small burst sizes, the two models are nearly identical. The two models differ more in very small communities and when burst sizes are very large, with the full model predicting somewhat higher levels of metabolite noise. This occurs because the growth feedback makes it more difficult for a “struggling” cell to correct its metabolite imbalances. An imbalanced cell has a lower growth rate, which makes it less likely to produce a corrective enzyme burst, thus leading to overall more persistent imbalances and increasing noise.

Degradation of metabolites: This assumption was included in the model to prevent unrealistic build-ups of metabolites within the cell. Without this assumption, a cell with poor regulation will build up massive, non-physical pools of one metabolite which it can later use. This does not reflect the reality of bacteria, which have a finite osmotic capacity and will release excess osmolytes (see Buda 2016 in PNAS). In addition, there also exist solubility limits that constrain metabolite storage. The assumption of degradation in the extracellular space was added to reflect the fact that in natural environments there may be other organisms present that consume free metabolites, or metabolites may simply diffuse away in 3D.

However, we agree that it would be informative to examine a model without degradation. We therefore explored such a model and present the results in a new appendix called “Metabolism models without metabolite degradation”. Many core behaviors of the model, such as the negative effect of burst size and the benefit of resource sharing, remain. However, the lack of degradation also introduces a counter-intuitive effect into the model: more metabolite storage can actually lead to an increased probability of death. This increased susceptibility to death is a consequence of large metabolite build-ups leading to sudden metabolite “crashes” that can kill the cell. We show the metabolite and enzyme timecourses of a cell undergoing this crash behavior in Appendix 9 – figure 10CD. At around t = 15, the cell develops an excess of the blue metabolite with a deficit of the pink metabolite, limiting its growth. In the time it takes the organism to produce a new burst of the pink enzyme, it continually builds up its pool of the blue metabolite, as there is no degradation term to limit the pool’s size. Eventually, at around t = 35, the cell produces a burst of the pink enzyme, generating more pink metabolite and allowing a resumption of steady growth. Since there is such a large build-up of the blue metabolite, it takes the cell a long time to deplete this metabolite pool. In this time, the cell produces many bursts of the pink enzyme, almost entirely replacing the cell's enzyme pool and leaving the cell with almost no blue enzyme. At around t = 55, the blue metabolite crashes, rapidly becoming the new limiting metabolite. At this point, the cell is left in a difficult position, suddenly having almost none of the now limiting blue metabolite and having an enzyme pool entirely dedicated to production of the pink metabolite. Growth rapidly ceases, as the remaining blue metabolite pool is insufficient to produce a corrective enzyme burst. Thus, paradoxically, allowing microbes to store larger pools of metabolites by removing degradation can actually make them more likely to die.

5. In lines 142-144 and 185-186 the authors talk about the role of extracellular volume/cell density; however, no explanation is given of how these results were derived. I think these statements need to be supported by either a SI figure or by an intuitive explanation of how r_v_ impacts model predictions.

We have now added a new SI figure (Appendix 9 – Figure 3) with simulations showing the effect of extracellular volume on growth rates. We provide more detailed explanations of the behavior and also include a comparison of the simulations with the analytically predicted maximum growth rate at infinite extracellular volume, showing they are in agreement.

6. In the analytical analysis presented in lines 187-205 the authors make use of various additional simplifying assumptions. Without additional details it is very hard to judge to what extend they affect the presented results. I think some additional discussion on the rational/effect of these assumptions would be highly beneficial to the reader (alternatively one could add a SI figure to e.g. compare the predictions of Equation 5 with data from simulations).

We have now added an SI figure (Appendix 9 – figure 4) comparing the predictions of the full model with the Langevin model. We have also modified the initial description of the Langevin model to make it clear that the primary modification is the removal of the nonlinear feedback between growth and enzyme dynamics.